# Photogated two conductive pathways of donor-acceptor Stenhouse adducts in single-molecule junctions

Fanxi Sun [1,6], Shengqing Jiang[1,6], Hanjun Zhang[1,2,6], Rui Wang[3] ✉, Yu Ji[1], Songjun Hou[4], Maolin Zhang[1], Gaolu Zhu[1], Tianfang Shi[1], Jiayu Li[1], Yuantao Zheng[1], Wenshu Liu[1], Yangyang Pan[1], Hao Luo[1], Xu Deng [5], Yonghao Zheng [1] ✉, Chen Wei[1] & Dongsheng Wang [1] ✉

Manipulating intramolecular electron transportation can fundamentally modulate the optical property, electromagnetic behavior and chemical reactivity of molecules. Achieving simultaneous control of multiple (≥2) transport pathways within a single molecule, however, remains a significant challenge. Herein, we report light-gated modulation of two distinct conductive pathways in single donor-acceptor Stenhouse adduct (DASA) molecules using the scanning tunneling microscopy break-junction (STM-BJ) technique. The donor and π-bridge pathways are separately controlled by designing DASAs with two thiomethyl anchoring sites. In the donor pathway, a side-chain modulation mechanism operates, where *linear*-to-*cyclic* isomerization induces electronic redistribution and increases the conductivity. In contrast, the π-bridge pathway is governed by a main-chain modulation mechanism, in which deformation of the π-conjugated backbone decreases the conductivity. By synthesizing DASAs containing three thiomethyl anchoring sites, these two conductive pathways are integrated within a single-molecule junction and can be simultaneously modulated under 635 nm red-light irradiation and dark relaxation. The π-bridge transport in the *linear* state exhibits mixed through-bond and through-space character, while photoisomerization leads to an increased through-space contribution in the *cyclic* state driven by cyclopentenone formation. These results highlight DASAs' potential in understanding molecular electronics and developing photoresponsive molecular-scale devices.

Intramolecular electron transport is an essential topic in molecular electronics, which is closely interrelated with the optical property, electromagnetic behavior and chemical reactivity of molecules[1–3]. Manipulating electronic conducting pathways not only enables a direct and fundamental understanding of molecular performance but also demonstrates the great potential for constructing molecular-scale devices[4–6]. Stimuli-responsive molecules with switchable chemical structure offer opportunities to in situ manipulate intramolecular

[1]School of Optoelectronic Science and Engineering, University of Electronic Science and Technology of China, Chengdu, China. [2]School of Pharmacy, Qujing University of Medicine & Health Sciences, Qujing, China. [3]School of Chemistry and Molecular Engineering, East China University of Science and Technology, Shanghai, China. [4]Department of Physics, Lancaster University, Lancaster, UK. [5]Institute of Fundamental and Frontier Sciences, University of Electronic Science and Technology of China, Chengdu, China. [6]These authors contributed equally: Fanxi Sun, Shengqing Jiang, and Hanjun Zhang. ✉e-mail: rui_wang@ecust.edu.cn; zhengyonghao@uestc.edu.cn; wangds@uestc.edu.cn

electron transportation[7,8]. Molecular-scale devices have been extensively constructed with dynamically and precisely regulated electrical property under the control of light[9–11], electric field[12–14], magnetic field[15,16], temperature[17,18] and chemical stimulation[19,20], enabling them to sense, respond to, and adapt to their environment.

Typically, light exhibits efficient and contactless features and is known as an ideal stimulus for the controlling of intramolecular electron transportation. Single-molecule conductance on photoresponsive molecules, including diarylethene[10,21], azobenzene[22,23] and spiropyran[24,25] has been investigated using techniques such as graphene-based electrodes[10,26], scanning tunneling microscopy break junction (STM-BJ)[22,23] and mechanically controllable break junction[21,24,27]. However, current photoresponsive molecular-scale devices face two major limitations. Firstly, photogating multiple conductive pathways within a single molecule remains largely unexplored. This significantly limits the functional complexity of molecular-scale devices, especially those requiring multi-state logic, directional transport control or pathway-specific signal gating[28]. Secondly, UV excitation ($\lambda < 380$ nm) is still required for at least one switching direction in many photoresponsive molecules used in single-molecule studies[29]. Irradiation in this short-wavelength regime can introduce undesired effects such as photodegradation[29], photoelectric responses[30], plasmon-related perturbations[31,32] or localized heating[33] in metal nanogaps, which may interfere with conductance measurements and compromise device stability.

Donor-acceptor Stenhouse adducts (DASAs) offer a promising solution to both limitations. Structurally, DASAs exhibit a typical chemical structure with a triene π-bridge located between an electron donor and an electron acceptor. The electron donor and acceptor supply multiple potential anchoring sites, allowing for the deliberate construction of distinct and addressable conductive pathways[34]. Functionally, DASAs respond to low-energy visible light (550–750 nm), which minimizes the interference with metal electrodes and ensures compatibility with conductance measurements[35,36]. These advantages position DASAs as a versatile platform for developing advanced photoresponsive molecular-scale devices with multiple and switchable conductive pathways. Previous research on DASAs is mostly focused on their isomerization mechanism[37–39] and structural design[34,40], as well as applications based on their photoresponsive behavior[41–43]. Nevertheless, studies on the light-controlled electrical behavior, particularly on the single-molecule level, are still lacking.

In this study, we reveal light-controlling of intramolecular electron transport through two conductive pathways in single DASAs molecules, where the molecular conductance is determined using the STM-BJ technique (Fig. 1a). A series of DASA derivatives bearing one, two, or three thiomethyl anchoring sites were synthesized by introducing thiomethyl groups onto the electron donor and/or acceptor units (Fig. 1c, d, Supplementary Figs. 1–8)[44,45]. A 635 nm laser was used to control the photoisomerization while minimizing interference with metal electrodes (Fig. 1a). Using DASAs with two thiomethyl anchoring sites, two distinct conductive pathways through the donor moiety and the conjugated π-bridge, termed the donor pathway and the π-bridge pathway respectively, are independently examined, revealing markedly different responses to light-induced structural changes (Fig. 1b).

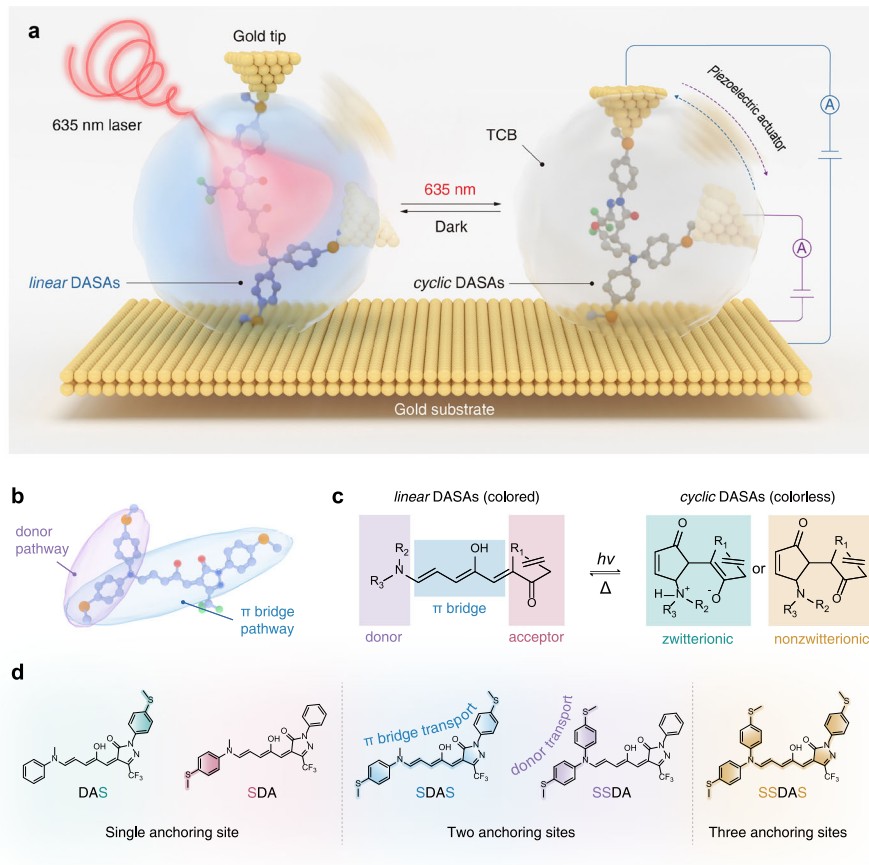

**Fig. 1 | Single-molecule conductance measurement setup and basic information of thiomethyl-anchored DASAs. a** Schematic illustration of the in-situ laser-irradiated single-molecule conductance measurement setup, DASAs isomerize between *linear* and *cyclic* in a 1,2,4-trichlorobenzene (TCB) droplet under the control of a 635 nm laser. **b** Schematic illustration of the donor pathway and π-bridge pathway in DASAs. **c** General chemical structure of *linear* and *cyclic* DASAs. **d** Chemical structures of DASAs with a single anchoring site (DAS and SDA), two anchoring sites (SDAS and SSDA), and three anchoring sites (SSDAS), a molecule with three anchoring sites enabling alternative two-terminal junction configurations.

Building on these results, the donor and π-bridge pathways are integrated within a DASA containing three thiomethyl anchoring sites, enabling their simultaneous modulation under red light irradiation and dark relaxation within a single-molecule junction. These results highlight the unique potential of DASAs as photoresponsive molecular platforms for controlling intramolecular electron transport, enabled by their intrinsic responsiveness to long-wavelength visible light and their modular molecular structure containing multiple anchoring sites.

## Results and discussion

### Working principles of photogated two conductive pathways in single-molecule junctions

The STM-BJ works as an STM tip, typically made of gold, being brought into close proximity to a gold substrate (Fig. 1a). The substrate is covered with a droplet of 1,2,4-trichlorobenzene (TCB) containing the testing molecules, which are often functionalized with anchoring groups, such as thiomethyl -SMe, amino -NH$_2$, pyridyl -Py and carboxyl -COOH[46,47]. While the tip is approaching the substrate, a metal-metal contact is initially formed, which is deformed during the retraction of the tip. During the retraction, a single molecule may bridge the gap between the tip and substrate, forming a molecular junction[48,49]. A 635 nm laser was positioned outside the main STM-BJ setup to minimize vibration and electrical noise generated by its cooling system and power-supply fans, and its output was delivered to the junction cell through a long optical fiber.

DASAs as the target molecules exhibit intriguing electronic property due to the unique modular structure, where a triene π-bridge locates between an electron donor and an electron acceptor, which results a push-pull nature with conjugation through the entire molecule (Fig. 1c). Visible light irradiation disrupts the D-π-A conjugated structure, converting the central triene π-bridge into a cyclopentenone ring (Fig. 1c). A series of third-generation of DASAs with -CF$_3$ modified acceptor were synthesized for the investigation of photogated electron transportation due to the red-light-triggered isomerization (Supplementary Figs. 1–8 and 56–78, see detail in the Supplementary section 1). DASAs with one, two, or three thiomethyl anchoring sites were synthesized by substituting thiomethyl groups onto the donor and acceptor units (Fig. 1d)[44,45]. Molecules containing a single thiomethyl anchoring site (DAS and SDA) are used to rule out parasitic conductance arising from non-thiomethyl contacts. Molecules with two anchoring sites enable two distinct photomodulated conductive pathways, allowing electron transport either through the donor unit (SSDA) or through the π-bridge (SDAS). Additionally, a molecule with three anchoring sites (SSDAS) integrates the donor and π-bridge pathways within a single molecular framework. Although the molecule contains three anchoring sites, at any given time only two are simultaneously coupled to the electrodes to form a two-terminal junction. The transition between the donor and the π-bridge pathways occurs during mechanical stretching, where one anchoring contact remains stable while the other anchoring contact is exchanged.

### Molecular orbital and photoisomerization property of DASAs

To gain deeper insights into how this modular architecture influences electron transport under light stimulation, we investigated the molecular orbital distribution (Supplementary Figs. 9–13, see detail in the Supplementary section 2) and photoisomerization behavior of two representative DASA derivatives, SSDA and SDAS, using density functional theory (DFT) calculations and time-resolved UV-vis spectroscopy (Fig. 2). These two molecules feature contrasting anchoring strategies: in SSDA, both anchors are fixed to the donor moiety, enabling electron transport primarily through the donor framework (main chain), while the extended triene π-bridge and acceptor serve as a photoactive side chain (Fig. 2a); in SDAS, the anchors are placed on both donor and acceptor ends, forcing electrons to traverse the entire π-conjugated system (main chain) (Fig. 2b).

Upon irradiation with 635 nm red light, both molecules undergo a *linear*-to-*cyclic* isomerization, transforming the triene linker into a cyclopentenone ring (Fig. 2a, b). This reaction disrupts the continuous π-conjugation and leads to a dramatic reorganization of the highest occupied molecular orbital (HOMO). In the *linear* form of SSDA, the HOMO is uniformly delocalized across the main chain and side chain, forming a conjugated extension that is electronically coupled to the triene π-bridge (Fig. 2a). Under red light irradiation, the *linear*-to-*cyclic* isomerization induces HOMO collapsing onto the donor pathway, with minimal orbital density remaining on the side chain. These indicate a side-chain modulation mechanism for the electron transport through the donor pathway. In contrast, for SDAS, the HOMO is distributed along the main conjugated backbone, and the changes in electron transport occur through a distinct main-chain modulation mechanism operating along the π-bridge (Fig. 2b). In its *linear* state, the HOMO shows a significant contribution over the triene π-bridge. Upon light-induced cyclization, the disruption of conjugation by the cyclopentenone unit leads to a more localized HOMO distribution on the donor side. This loss of delocalization predicts a significant disruption to through-bridge electron transport.

These theoretical results are further corroborated by time-resolved UV-vis absorption spectra (Supplementary Figs. 15–19, Supplementary Tables 1–35, see detail in the Supplementary section 3), which reveal a rapid and characteristic color change upon red-light irradiation (Fig. 2c). In the *linear* state, SDAS display strong absorption bands in the visible light region ($\lambda_{max} = 636$ nm), arising from the extended D-π-A conjugation of their triene bridges (Fig. 2c, Supplementary Fig. 55). Upon continuous 635 nm light exposure, these absorption bands steadily decrease in intensity, reflecting the isomerization of the *linear* triene to a *cyclic* cyclopentenone ring that interrupts π-conjugation. The spectra evolve toward a photostationary state within tens of seconds, where the absorption plateau signifies that most molecules have transitioned to the *cyclic* form (Fig. 2c). Under dark conditions at 22 °C, the *cyclic* SDAS spontaneously and rapidly isomerize to the colored *linear* form, with a first-order kinetic fitting[50,51] yielding a characteristic relaxation time of 24.35 s (Fig. 2d, e). Furthermore, three photoswitches (SSDA, SDAS, SSDAS) display excellent fatigue resistance, with their main absorption bands showing only minimal loss after multiple photoswitching cycles (Supplementary Figs. 20–22). Moreover, the substitution of thiomethylic anchoring groups on the donor and/or acceptor moieties significantly influences the fundamental physicochemical properties of DASAs, including absorption wavelength, full width at half maximum (FWHM) and molar extinction coefficient (Supplementary Fig. 55c, d, Supplementary Fig. 23)[52–56]. The isomerization behavior, such as the transition state energy barrier and the kinetics of thermal back-isomerization, is demonstrated to be closely interrelated with the number of introduced thiomethylic groups (Fig. 2e, Supplementary Fig. 55a, b, d). The interplay between these factors might be originate from the difference in the molecular polarity index (*MPI*) of the DASAs, which is comprehensively discussed in Supplementary Fig. 55 (Supplementary Fig. 14).

These pronounced changes in visible absorbance align with the theoretical prediction of orbital localization and confirm the effective disruption of conjugation, serving as a direct optical signature of the light-induced structural modulation. Taken together, the evolution of the frontier orbitals and absorption properties provides a mechanistic basis for the photogated conductance modulation explored in the following sections.

### Light-controlling the donor pathway conductance for SSDA

We first examine the light-controlled conductance through the donor pathway, which is supplied by the structurally fixed diphenylamine unit on SSDA. In this case, the *linear-cyclic* isomerization induces significant changes of the conjugated structure on the side chain, working as a "side-coupled quantum dot"[57,58] and regulating the electronic

 

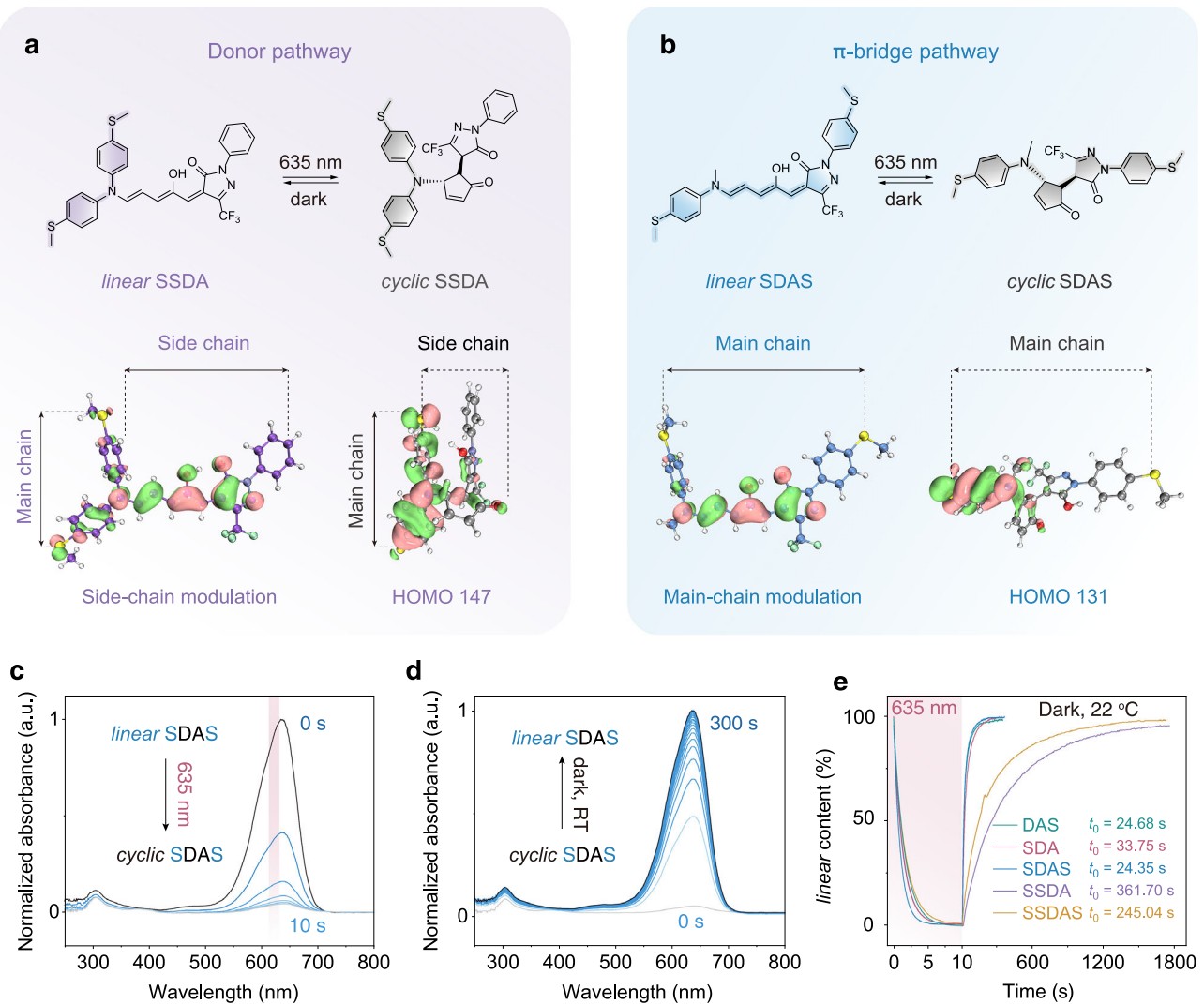

**Fig. 2 | Molecular orbital and photoisomerization behavior of DASAs with different conductive pathways. a**, **b** Molecular structures and calculated highest occupied molecular orbital (HOMO) distributions of (**a**) SSDA and **b** SDAS in their *linear* and *cyclic* isomeric forms. The side chain and main chain on the conductive pathways are marked. **c**, **d** Time-resolved UV-vis absorption spectra of SDAS (0.1 mM in TCB) (**c**) under 635 nm light irradiation and **d** in the dark. **e** Kinetics of *linear*-to-*cyclic* (left, under 635 nm light, 1 mW/cm²) and *cyclic*-to-*linear* (right, thermal relaxation at 22 °C) isomerization processes.

conductance on the main transport pathway (Fig. 3a). The conductance of *linear* and *cyclic* SSDA (0.1 mM in TCB) was measured without data selection at the bias voltages of 100 mV, 300 mV and 500 mV under dark and 635 nm laser irradiation, respectively (Supplementary Figs. 31–36). More than a thousand traces were recorded to obtain the 2D conductance histogram without any data selection (Fig. 3b, c). 2D conductance histograms reveal two distinct conductance bands in the range of $-4.5 \leq \log (G/G_0) \leq -2.5$ for both the without-irradiation and under-irradiation measurements (Fig. 3b, c). The lengths of these molecular plateaus align closely with the electrode-molecule distances predicted by DFT calculations for both the *linear* and *cyclic* isomers (Supplementary Fig. 48).

At a bias voltage of 300 mV, the conductance values of the *linear* and *cyclic* SSDA are $10^{-3.91}$ $G_0$ and $10^{-3.71}$ $G_0$, respectively. To investigate the data robustness, we performed two additional "light-off/light-on" cycles (Supplementary Fig. 35) and conducted orthogonal measurements at two other bias voltages (100 mV and 500 mV, Supplementary Figs. 31–33), resulting in a total of five independent data sets. Under 300 mV, the *linear* isomer yielded conductance values of $10^{-3.92}$ $G_0$ and $10^{-3.88}$ $G_0$ after the following two isomerization cycles, while the *cyclic* isomer gave $10^{-3.60}$ $G_0$ and $10^{-3.64}$ $G_0$ (Supplementary Fig. 35). At 100 mV

and 500 mV, the *linear*-state conductance appears at $10^{-3.61}$ $G_0$ and $10^{-3.65}$ $G_0$, respectively, whereas the *cyclic*-state conductance appears at $10^{-3.39}$ $G_0$ and $10^{-3.46}$ $G_0$ (Supplementary Figs. 31–33).

Variations in bias voltage may influence the relative alignment of molecular orbitals with the electrode Fermi level and can lead to conductance fluctuations[59,60], but the focus here is on the reproducibility of the isomer-dependent difference. Across all five measurements at 100, 300, and 500 mV, the data consistently show that *linear-cyclic* isomerization produces a small but reproducible conductance enhancement of approximately 0.2 log units (a factor of about 1.5), as summarized in Supplementary Fig. 36. The transmission spectra also show a similar trend: the transmission peak near the Fermi energy level shifts slightly from *linear* SSDA ($10^{-2.90}$ $G_0$) to *cyclic* SSDA ($10^{-2.73}$ $G_0$) (Fig. 3f)[61,62]. This observation is supported by the calculated frontier orbital distributions, where in both isomers, the HOMO remains the dominant transport orbital (Fig. 2a). As a result, the conductance pathway through the donor remains largely intact, with only a slight enhancement in transmission efficiency.

To further investigate the electron transport mechanism in different isomers, we performed flicker noise measurements on the conductance plateaus of *linear* and *cyclic* SSDA (Supplementary

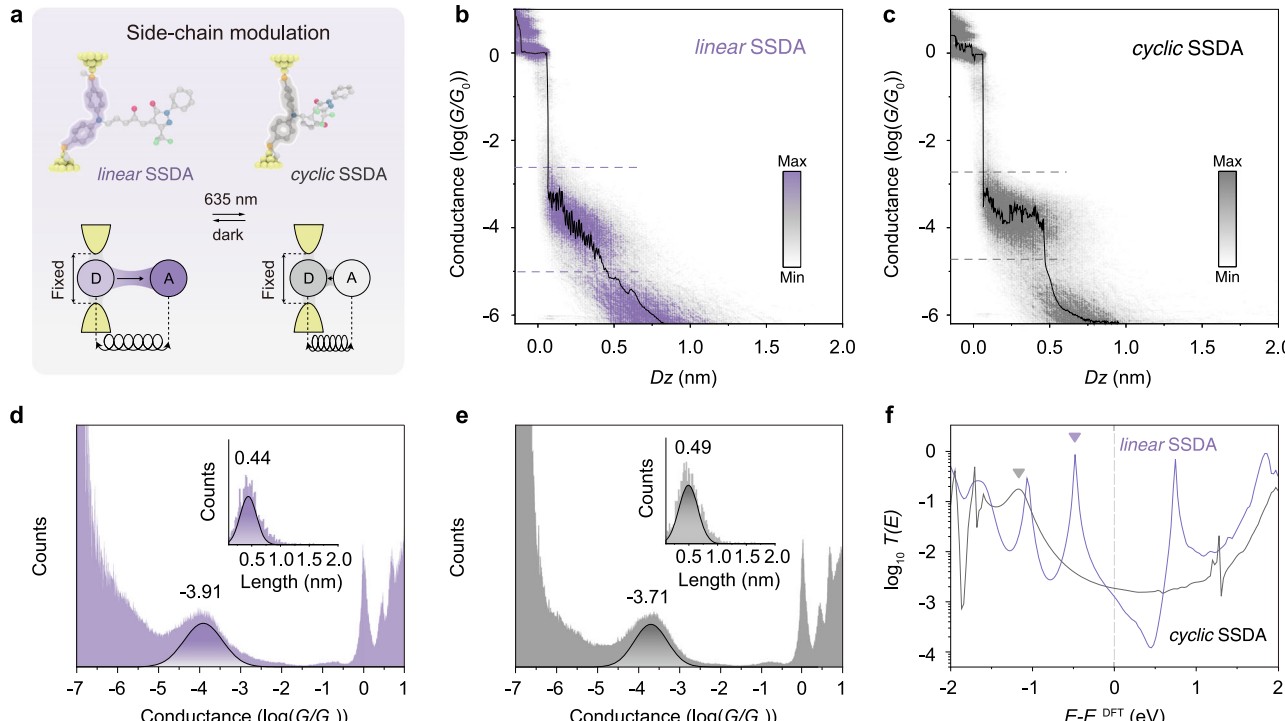

**Fig. 3 | Controlling the conductance of donor pathway. a** Schematic illustration of controlling the electron transport through the donor pathway by the *linear-cyclic* isomerization of SSDA, which is based on a side-chain modulation mechanism. **b**, **c** 2D conductance-distance histograms of (**b**) *linear* SSDA and **c** *cyclic* SSDA at a bias voltage of 300 mV, the color bar indicates the number of counts. **d** One-dimensional logarithmic conductance histograms of *linear* SSDA at bias voltages of 300 mV. Inset shows the relative stretching distance distributions of the *linear* SSDA junction. **e** One-dimensional logarithmic conductance histograms of *cyclic* SSDA at bias voltages of 300 mV. Inset shows the relative stretching distance distributions of the *cyclic* SSDA junction. **f** The theoretical calculated transmission curves[61,62] of *linear* SSDA and *cyclic* SSDA.

Fig. 34). The STM tip was suspended for 150 ms once a conductance plateau was detected. Thousands of traces were analyzed to extract the flicker noise power spectral density (PSD), which was normalized by the average conductance ($G$) of each plateau. For both *linear* and *cyclic* SSDA, the noise power scales with conductance following a power-law dependence, with extracted exponents of $G^{1.51}$ and $G^{1.61}$ for the *linear* and *cyclic* isomers, respectively (Supplementary Fig. 34). According to previous reports[63], a PSD exponent close to 1.0 indicates a dominant through-bond transport mechanism, whereas values approaching 2.0 suggest increasing contributions from through-space tunneling. This interpretation is supported by transmission pathway calculation, which reveals that both isomers exhibit mixed through-bond (blue arrows) and through-space (red arrows) transport channels (Supplementary Fig. 51). Therefore, the intermediate exponents (1.5–1.6) observed experimentally reflect a combined contribution of the two mechanisms, which do not exhibit significant differences between *linear* and *cyclic* SSDA.

The example of SSDA demonstrates a device strategy in which the conductance of the fixed main chain can be modulated by the conjugated structure variation on the side chain, enabled through rational alignment of the molecular orbital and anchoring site. Moreover, the opposite shifts of the two conductance features upon photoisomerization highlight the distinct ways in which different parts of the molecule respond to light, offering a foundation for exploring two-channel conductance mechanisms in single-molecule junctions.

**Light-controlling the π-bridge pathway conductance for SDAS**
Building on the insights obtained from the side-chain modulation in SSDA, we next examine a distinct modulation strategy in SDAS, where electron transport necessarily traverses the entire molecular backbone (Fig. 4a). Analysis of the transmission pathways (Supplementary

Fig. 52) reveals that electron transport through the π-bridge in *linear* SDAS follows a through-bond transport mechanism. However, once photoisomerization drives the molecule into the cyclopentenone form at the photostationary state, charge transport is significantly hindered and dominated by through-space interactions (Supplementary Fig. 52). This shift transforms the conduction mode from a "resistive regime" in the *linear* state to a "capacitive regime" in the *cyclic* state. Therefore, the light-controlling of the conductance of the π-bridge pathway relies on the main-chain modulation mechanism (Fig. 4a).

For the *linear* SDAS, STM-BJ conductance histograms show a pronounced peak at $10^{-5.13}$ $G_0$ (Fig. 4b, d, Supplementary Figs. 39, 40). Upon light irradiation, this peak is dramatically suppressed to $10^{-5.75}$ $G_0$, signifying that direct π-bridge conductance is largely "turned off" by 0.62 orders of magnitude (Fig. 4c, e). Across the five orthogonal measurements, including two additional "light-off/light-on" cycles at 300 mV (Supplementary Fig. 37–38) and measurements at 100 mV and 500 mV (Supplementary Figs. 39, 40), the conductance of both the *linear* and *cyclic* SDAS remains stable. The *linear* state fluctuates slightly around $10^{-5.15}$ $G_0$, while the *cyclic* state remains near $10^{-5.70}$ $G_0$, indicating only minor variations across all applied bias voltages. Moreover, the change in molecular junction length induced by isomerization is also validated by single-molecule conductance measurements. The apparent molecular length decreased from 1.60 nm for *linear* SDAS (including the ~0.5 nm gold-gold snap-back distance) to 1.22 nm for *cyclic* SDAS (Fig. 4b), in good agreement with the DFT-calculated junction lengths of 15.18 Å and 11.86 Å for the respective isomers (Supplementary Fig. 47). The downward trend in conductance upon photoisomerization is further corroborated by DFT-calculated transmission spectra, where the transmission near the Fermi level decreases from $^{-5.43}$ $G_0$ (*linear* SDAS) to $^{-6.16}$ $G_0$ (*cyclic* SDAS), increasing the energy offset between the Fermi level, thus lowering the tunneling probability (Fig. 4f).

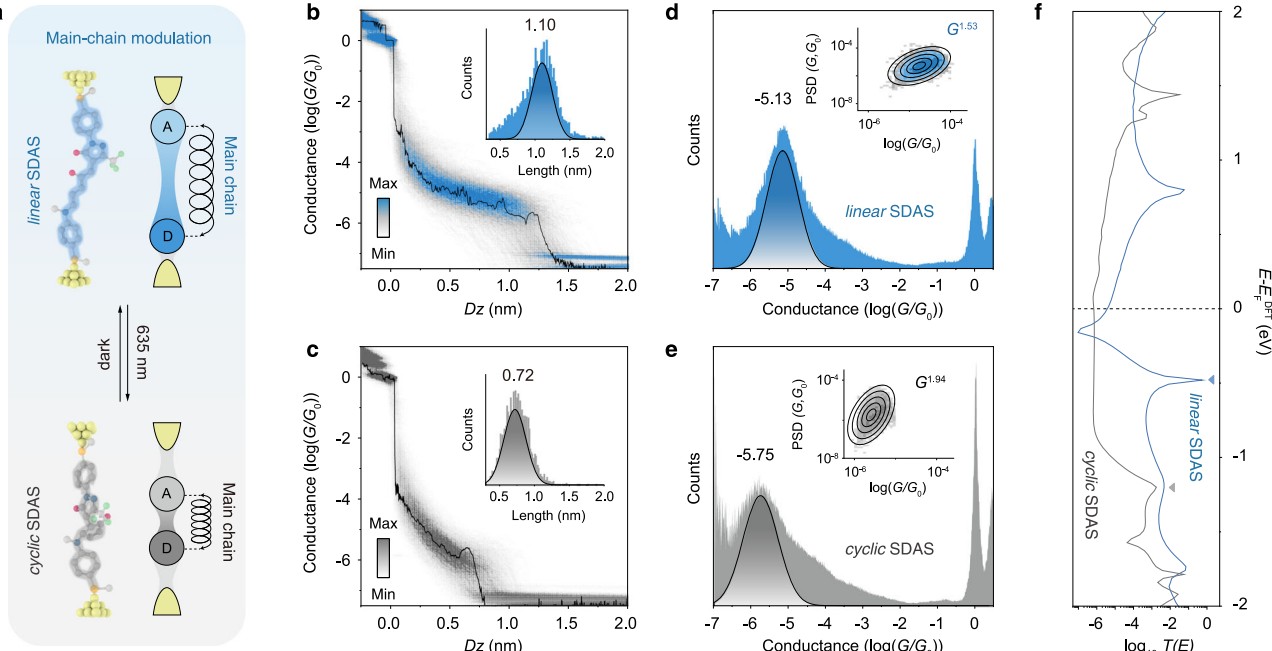

**Fig. 4 | Controlling the conductance of π-bridge pathway. a** Schematic illustration of controlling the electron transport through the π-bridge pathway by the *linear-cyclic* isomerization of SDAS, which is based on a main-chain modulation mechanism. **b, c** 2D conductance-distance histograms of (**b**) *linear* SDAS and **c** *cyclic* SDAS at a bias voltage of 300 mV, the color bar indicates the number of counts. The inset shows the relative stretching distance distributions of *linear* SDAS

and *cyclic* SDAS junctions. **d** One-dimensional logarithmic conductance histograms of *linear* SDAS at bias voltages of 300 mV. The inset shows the flicker noise PSD of *linear* SDAS. **e** One-dimensional logarithmic conductance histograms of *cyclic* SDAS at bias voltages of 300 mV. The inset shows the flicker noise PSD of *cyclic* SDAS. **f** The theoretical calculated transmission curves of *linear* SDAS and *cyclic* SDAS.

Flicker noise analysis further highlights the mechanistic shift beyond the conductance change. The noise power scaling exponent increases from 1.53 in the *linear* SDAS to 1.94 in the *cyclic* SDAS (Fig. 4d, e). Specifically, the *linear* SDAS exhibits noise characteristics consistent with mixed through-bond and through-space transport, where a dominant through-bond pathway is supplemented by through-space coupling. In contrast, photoisomerization to the *cyclic* form leads to a substantial suppression of conductance, accompanied by an enhanced contribution from through-space interactions and weakened through-bond transport. Transmission pathway analysis offers a more intuitive visualization of the electron transport mode shift across the π-bridge in the *linear* and *cyclic* isomers (Supplementary Fig. 52). In the *linear* SDAS, electron transport exhibits a mixed through-bond and through-space character: the central triene π-bridge provides an efficient through-bond pathway, while localized through-space coupling is present at the pyrazole unit on the acceptor side (highlighted by red arrows in Supplementary Fig. 52). By contrast, in the *cyclic* SDAS, deformation of the extended π-conjugation leads to a reduced through-bond contribution, accompanied by an increased density of through-space coupling pathways between the pyrazole ring and the newly formed cyclopentenone moiety, indicating that electron transport is dominated by through-space tunneling (Supplementary Fig. 52). The junction thus evolves toward a "capacitive regime", where electron flow primarily relies on non-bonded electrostatic coupling and continuous covalent pathways are largely lost.

A comparison with SSDA further clarifies how photoisomerization differs between these two systems. For SSDA, both the isomers preserve the same donor-side benzene-N-benzene transport pathway, and HOMO maps (Fig. 2a, b) show that the *cyclic* form exhibits a more concentrated orbital distribution on this fixed donor segment compared to the *linear* form. This redistribution slightly strengthens the electronic coupling while keeping the backbone topology essentially unchanged, which accounts for the modest conductance enhancement. In contrast, photoisomerization directly reshapes the π-bridge

of SDAS. In the *linear* isomer, the HOMO shows a significant contribution to the extended π-system. After isomerized to the *cyclic* form, the π-bridge is disrupted, and the HOMO becomes more localized, leading to reduced conjugation and lowered conductance.

Moreover, to assess the possibility of anchoring through atoms other than the thiomethyl group, two molecules with a single thiomethyl anchoring site (DAS and SDA) were designed (Fig. 1d). Under the same conditions with multiple biases, neither of these two molecules exhibits conductance features similar to those of SSDA and SDAS (the issue of unconventional anchoring is discussed in detail in section 4.2 of the SI, Supplementary Figs. 24–30). While the exact atomic-scale contact geometry in STM-BJ measurements cannot be uniquely determined, these control experiments indicate that contributions from non-thiomethyl anchoring, including possible weaker interactions involving C, N, or O atoms, if present, do not play a dominant role in the conductance features discussed above.

### Photogated two conductive pathways in SSDAS junctions

While electron transport through either the donor pathway or the π-bridge pathway can be modulated by the *linear-cyclic* isomerization of SSDA and SDAS, respectively, we further investigate the integration of these two conductive pathways within a single-molecule junction by using SSDAS containing three thiomethyl anchoring sites (Fig. 1d). This dual-channel design allows us to simultaneously probe the donor and π-bridge contributions to conductance under identical conditions.

As shown in the 2D conductance-displacement histograms, both *linear* and *cyclic* SSDAS display two distinct conductance plateaus, corresponding to the donor and π-bridge pathways (Fig. 5a, b). When the SSDAS is in the *linear* form, the high-conductance plateau (P1) reflects electron transport through the donor pathway, while the lower-conductance plateau (P2) originates from transport across the π-bridge pathway (Fig. 5a). Upon light-induced *linear*-to-*cyclic* isomerization, two corresponding features (P3 and P4) become more clearly distinguishable, indicating that both pathways persist and can

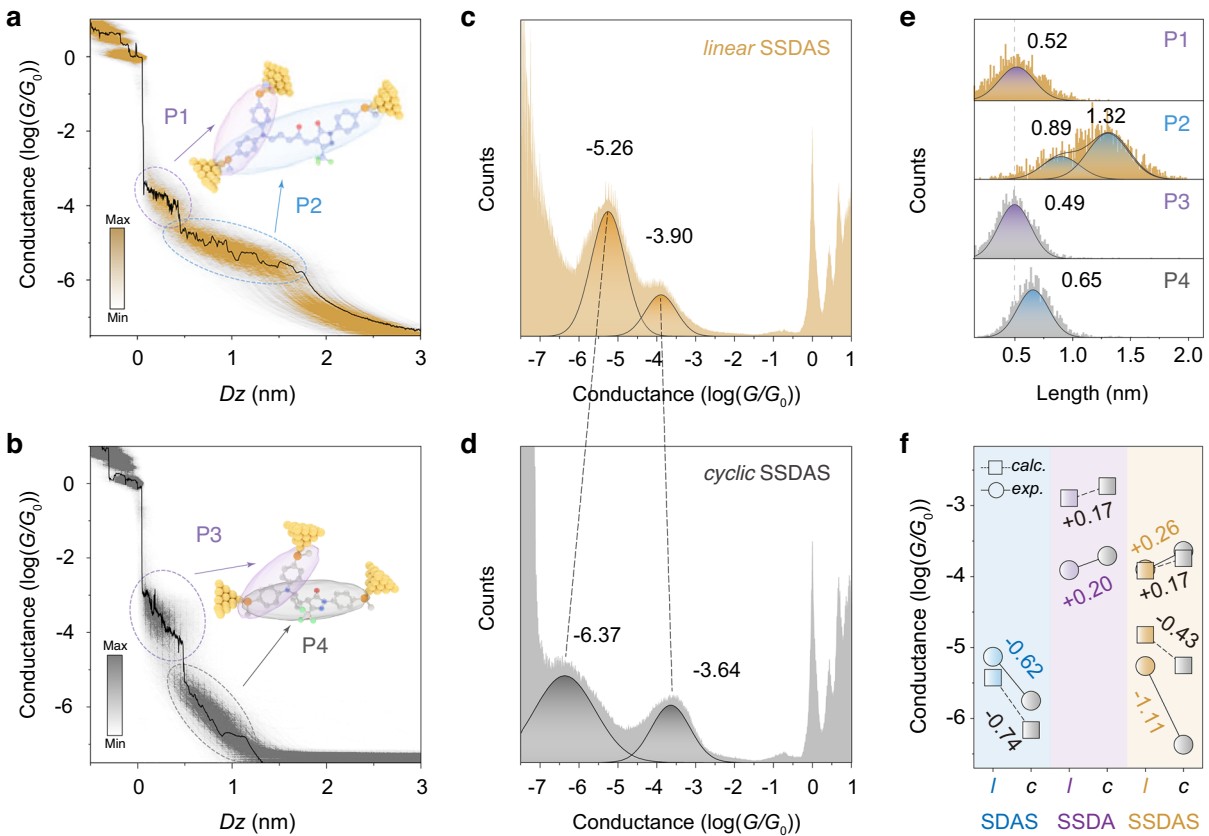

**Fig. 5 | Photogated the donor and π-bridge pathways in single-molecule junctions. a, b** 2D conductance-distance histograms of (**a**) *linear* SSDAS and **b** *cyclic* SSDAS at a bias voltage of 300 mV, the color bar indicates the number of counts. **c, d** One-dimensional logarithmic conductance histograms of (**c**) *linear* SSDAS and **d** *cyclic* SSDAS at a bias voltage of 300 mV. **e** The relative stretching distance distributions of *linear* SSDAS and *cyclic* SSDAS junctions at a bias voltage of 300 mV. **f** Summary of calculated (the position near zero energy) and measured conductance changes for SDAS, SSDA and SSDAS before and under red light irradiation.

be independently resolved in the *cyclic* SSDAS (Fig. 5b). The plateau lengths extracted from single-molecule conductance traces are consistent with those observed in SSDA and SDAS, thereby validating the assignment of conductance features to the respective pathways (Fig. 5c). Notably, due to the presence of three thiomethyl anchoring sites in SSDAS, the π-bridge pathway can adopt two distinct binding configurations, resulting in two junction lengths of 1.39 nm and 1.82 nm (Fig. 5e, including the -0.5 nm gold-gold snap-back distance). These lengths closely match the DFT-calculated electrode-to-electrode distances of 15.0 Å and 20.3 Å, and are slightly longer than the 1.60 nm single dominant configuration observed in SDAS (Fig. 4f, Supplementary Figs. 49, 50).

The 1D conductance histograms reveal a pronounced bidirectional conductance switching upon photoisomerization (Fig. 5c, d, Supplementary Fig. 45). For the π-bridge pathway, the conductance drops from $10^{-5.26}$ $G_0$ to $10^{-6.37}$ $G_0$ at a bias voltage of 300 mV, corresponding to a 1.11-order-of-magnitude decrease (P2 to P4). By contrast, the donor pathway conductance increases modestly from $10^{-3.90}$ $G_0$ to $10^{-3.64}$ $G_0$ (P1 to P3). The conductance of *linear* and *cyclic* SSDAS remains stable across the five orthogonal measurements (Supplementary Figs. 41–45). This represents the first example of a single-molecule system where light stimulus induces opposite conductance responses in two independently addressable channels (Fig. 5c, d). Moreover, the conductance contrast between the two pathways is significantly amplified upon photoisomerization: the difference between the high and low conductance increases from 1.36 orders of magnitude in *linear* SSDAS (P1 vs. P2) to 2.73 orders in *cyclic* SSDAS (P3

vs. P4) (Fig. 5c, d). Further analysis of individual conductance traces confirms that both plateau types can frequently coexist within a single pulling event, including traces that sequentially display high-to-low transitions (P1 to P2 and P3 to P4), offering direct evidence of junction reconfiguration between donor and π-bridge paths during molecular elongation (Supplementary Fig. 46).

For these three molecules, SSDA, SDAS and SSDAS, the trends of conductance change upon light irradiation are consistently reproduced by DFT-calculated transmission values near the Fermi level (Fig. 5f, Supplementary Figs. 49, 50, 53, 54). Notably, the integrated pathways in SSDAS show more pronounced variations induced by the *linear-cyclic* isomerization compared to the individual ones in SSDA and SDAS. For the π-bridge pathway, the conductance decreases upon light irradiation in SSDAS is nearly twice that observed in SDAS (1.11 vs. 0.62 orders), while the enhancement of the donor pathway in SSDAS only slightly exceeds that of SSDA (0.26 vs. 0.20 orders) (Fig. 5f). This increased responsiveness is due to the greater redistribution of electrons within the molecule and the deformation of π-conjugation in the *cyclic* form (Supplementary Fig. 13).

In summary, this study offers an opportunity to precisely control the molecular electronic properties by photogating two conductive pathways in single-molecule junctions, where the electron transportation through the photoresponsive molecules can be switched and modulated by light. A series of DASAs with thiomethyl anchoring sites was synthesized for the construction of single-molecule junctions, including molecules containing one, two, or three thiomethyl anchoring sites (DAS, SDA, SSDA, SDAS, and SSDAS). First, electron

transport through the donor and π-bridge pathways is separately realized using DASAs with two thiomethyl anchoring sites. In SSDA, the donor pathway is realized, where electrons traverse the diphenylaminic donor and *linear-cyclic* isomerization modulates the conjugated structure on the side chain, thereby regulating the conductance of the main chain within 0.20 orders of magnitude. In contrast, the π-bridge pathway is realized in SDAS, where electrons traverse the entire molecule, including the donor, π-bridge, and acceptor units. Photoisomerization induces formation and deformation of the π-bridge along the main chain, leading to a significant redistribution between through-bond and through-space contributions: the *linear* state exhibits mixed through-bond/through-space transport, while the *cyclic* state shows an enhanced contribution from through-space transport, resulting in a conductance variation of 0.62 orders of magnitude. The donor and π bridge pathways are integrated into a single molecule junction using SSDAS, containing three thiomethyl anchoring sites, and can be simultaneously modulated under 635 nm red light irradiation, demonstrating photogated control of two conductive pathways at the single-molecule level.

These results establish a molecular platform in which multiple electron-transport modes can be selectively addressed by light, offering opportunities for multi-level conductance states and optically gated logic behaviors at the molecular scale. Such capabilities provide a conceptual foundation for the design of adaptive and stimuli-responsive molecular electronic elements. However, challenges remain in achieving precise control over electrode-molecular junction geometries and scalable integration into device platforms. Future work combining in situ spectroscopic characterization, advanced simulation and multi-functional junction architectures will be critical to fully harness the potential of photoresponsive molecules in nanoelectronics.

# Methods

### Materials
All the chemicals and reagents were used without further purification. 4-(Methylmercapto)aniline ($C_7H_9NS$, CAS No. 104-96-1) and ethyl 4,4,4-trifluoroacetoacetate ($C_6H_7F_3O_3$, CAS No. 372-31-6) were purchased from Meryer Chemical Technology Co., Ltd. (China). Hydrochloric acid (HCl, CAS No. 7647-01-0), dichloromethane ($CH_2Cl_2$, CAS No. 75-09-2), iodomethane ($CH_3I$, CAS No. 74-88-4), sodium hydride (NaH, CAS No. 7646-69-7), hexane ($C_6H_{14}$, CAS No. 110-54-3), tetrahydrofuran (THF) ($C_4H_8O$, CAS No. 109-99-9), diethyl ether ($C_4H_{10}O$, CAS No. 60-29-7), ethyl acetate ($C_4H_8O_2$, CAS No. 141-78-6), and sodium sulfate anhydrous ($Na_2O_4S$, CAS No. 15124-09-1) were purchased from Chengdu Kelong Chemical Co., Ltd. (China). Sodium nitrite ($NaNO_2$, CAS No. 7632-00-0), acetic acid ($C_2H_4O_2$, CAS No. 64-19-7), furfural ($C_5H_4O_2$, CAS No. 98-01-1), and 1,1,1,3,3,3-hexafluoro-2-propanol ($C_3H_2F_6O$, CAS No. 920-66-1) were purchased from Aladdin Chemicals (China). Stannous chloride ($SnCl_2$, CAS No. 7772-99-8), N-methyl-4-(methylsulfanyl)aniline ($C_8H_{11}NS$, CAS No. 58259-33-9), sodium hydroxide (NaOH, CAS No. 1310-73-2), and 3-trifluoromethyl-1-phenyl-1H-5-pyrazolone ($C_{10}H_7F_3N_2O$, CAS No. 321-07-3) were purchased from Macklin Biochemical Co., Ltd. (China). 4,4'-Dimethyl-thiodiphenylamine ($C_{14}H_{15}NS_2$, CAS No. 1310458-10-6) was purchased from Weidi Chemical Technology Co., Ltd. (China). 1,2,4-Trichlorobenzene ($C_6H_3Cl_3$, CAS No. 120-82-1) was purchased from Sigma-Aldrich Trading Co., Ltd. (China). Milli-Q water (resistivity: 18.2 MΩ × cm) was used throughout the project.

### Characterization
UV-vis absorption spectra were recorded on a Shimadzu UV-2600 spectrophotometer. The light-induced reaction kinetics were studied using a Lightway PQY-01 photochemical reaction evaluation system. Nuclear magnetic resonance (NMR) spectra were collected on a Bruker AVANCE III HD 400 MHz spectrometer to analyze molecular structures. ESI-HR-TOF-MS data were collected on a Bruker high-resolution TOF mass spectrometer. Single-molecule conductance measurements were performed using a STM-BJ-4.0 platform (XTech) based on the scanning tunneling microscopy-break junction (STM-BJ) technique. A programmable DC power supply (MS305D, Maisheng) was used to power the laser source. A 635 nm continuous-wave laser (K635EWDFN-5.000 W, BWT Beijing) was used as the monochromatic light source for photophysical studies. The light intensity and distance of the laser were recorded on a PCPlug V3 (LaserPoint).

# Data availability
Source data are provided with this paper.

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

## Acknowledgments

This work was supported by the National Natural Science Foundation of China (22405088 (R. W.), 52203134 (D. W.), 62375041 (C.W.), and

22375029 (Y. Z.)) and the Foundation of Science & Technology Department of Sichuan Province (2023ZYD0037 (Y. Z.), 2024YFHZ0307 (Y. Z.), and 2024NSFSC0249 (D. W.)).

## Author contributions

F.S., S.J., and H.Z. contributed equally to this work and performed all experimental investigations. F.S. and D.W. led the manuscript preparation and data visualization. R.W., S.H., and Y.J. carried out and supervised the transmission spectra and transmission pathway simulations. M.Z., G.Z., J.L., Y.Z., and T.S. performed the DFT calculations. C.W. and W.L. built the laser setup. Y.P. and H.L. synthesized the molecules. X.D., Y.Z., and D.W. initiated and supervised all aspects of the research.

## Competing interests

The authors declare no competing interests.
