## [Transparent Peer Review file · Nature Communications]

Photogated Two Conductive Pathways of Donor-Acceptor Stenhouse Adducts in Single-Molecule Junctions

Corresponding Author: Professor Dongsheng Wang

Version 1:

Reviewer comments:

Reviewer #1

(Remarks to the Author)

This is a highly interesting paper on how to modulate molecular conductance of a triple-anchored molecular photoswitch in a junction. The conductance measurements are very systematic and seem to be done excellently with proper reference measurements, but I should emphasize that I am not a physicist and not able to judge whether any additional test measurements are necessary for the conclusions. I can only say that I find the results very exciting and clearly presented, and I think this work should have a high impact. I only have one issue that I think needs to be addressed before publication. All new compounds must be fully characterized by $^1\text{H-NMR}$, $^{13}\text{C-NMR}$ and HRMS or EA. In particular, for the final MeS-functionalized Stenhouse adducts, high-resolution mass spectra or elemental analysis should be provided to meet the highest standards of the synthesis community. A minor comment, the authors list molecular photoswitches that have been studied in junctions. Also dihydroazulene could be added to this list (e.g., Nat. Commun. 2017, 8, 15436).

Reviewer #2

(Remarks to the Author)

In this article, the authors demonstrated light-controlled intramolecular electron transport through multiple conductive pathways in single DASA molecules using the STM-BJ technique. The results highlight the unique potential of DASAs as molecular transistor components, owing to their intrinsic responsiveness to near-infrared light and modular structure that enables multi-anchor configurations. I recommend this work for publication after revision. Several comments and suggestions are provided below:

1. Error bars should be added to Figure 3g to improve data reliability.
2. The authors achieved photoconversion from linear to cyclic molecular structures upon light irradiation, with reversion to the linear form after the light is turned off. Is this photoresponse fully reversible? If an irreversible component exists, what is its approximate proportion? Furthermore, can the reversible switching between linear and cyclic forms be sustained over multiple cycles? I suggest a repeatedly isomerization experiment should be done. Does repeated photochemical switching lead to molecular degradation or loss of functionality?
3. The simulation section in Figure 6 should clearly state the input parameters and the rationale behind the equivalent resistance calculations. It is recommended to include these details in the main text or the Supplementary Information.
4. Physical quantities in the figures and main text should be italicized.
5. The abstract is somewhat lengthy and could be condensed to more effectively emphasize the innovative aspects and key findings.
6. Although schematics are provided comparing "side-chain" and "main-chain" modulation mechanisms for SSDA and SDAS, adding a graphical abstract would help visually summarize the light-controlled dual-channel switching mechanism in SSDAS.
7. The "BJT-like" and "MOS-like" analogies are creative but should be used cautiously and accompanied by detailed explanations to avoid oversimplifying the differences between molecular-scale devices and macroscopic electronic components.
8. The conclusion could be strengthened by further emphasizing the potential implications of this work for the development of multi-state logic devices or optoelectronic intelligent materials.
9. The x-axis of the inset figures of typical individual conductance-distance traces in Figure 3d and 3e should be added.
10. It is better to have some discussions about why the conductance of cyclic-SSDA/SDAS is larger than that of linear-SSDA/SDAS. Since the length of cyclic type molecule is even longer than that of linear type, and the main tunneling route of

those molecules are the same of benzene-N-benzene, what's the determining point of Nitrogen atom to affecting the conductance? Electronegativity or anything else?

Reviewer #3

(Remarks to the Author)

Referee report about ms. Photogated Multiple Conductive Pathways of Donor-Acceptor Stenhouse Adducts in Single-Molecule Junctions by Fanxi Sun et al.

The manuscript reports the fabrication and characterization of phototunable donor acceptor Stenhouse-based molecules as well as STM breakjunction measurements and accompanying DFT calculations. The authors claim that some of these molecules may act as equivalents to transistors.

The manuscript contains major flaws and cannot be published.

The problems start already in the introduction where the authors claim that the plasma resonances of typical metals used for contacting single molecules would be in the UV range. This is simply wrong as a simple literature survey shows. They also claim that other phototunable molecules would be addressable in the UV range only. This is wrong as well. The most important classes of photochromic molecules with two distinct states can be activated also in the visible range (for one of the switching directions). They mention several drawbacks of activation in the UV and miss to mention the most important one of them: photodegradation.

The Stenhouse molecules studied here are activated in the visible range, but apparently this state is not stable since they relax back in the dark within a timescale of a few hundred seconds. This is a major drawback for their usage. It also puts the results of the transport measurements in question, since they were performed in the dark and no time frame is indicated. So, it cannot be excluded that the molecules were already partly relaxed back. In the bottom of page 4 the authors mention near infrared activation, which is not investigated here.

The authors repeatedly write about "multiple conductance pathways". In fact they discuss 2 (in the molecule with three anchoring sites), not multiple, and even this claim is not supported by evidence, because it is not made clear how these three anchoring points can be realized in the junction. Fig 1 a suggests three anchoring points and somehow also that the anchoring point might be changed by the piezo? This is all highly suggestive but not explained.

On page 8 the authors try to explain the change of the conductance of the main chain upon photocyclisation. They claim that in the linear state the HOMO would be delocalized while it would be localized in the cyclic form. In fact, Fig. 2b clearly shows that the HOMO is NOT extended over the whole linear chain. It is also interrupted almost in the middle of the molecule. In the cyclic form the HOMO is probably a bit more localized, but the molecule is also much shorter.

For the single-anchored molecules: How do the authors know where the top contacts the molecule?

About "V shape" dependence of the conductance on the bias voltage: There are only three data points: 100 mV, 300 mV, 500 mV. It is exaggerated to talk about V shape. It seems to be non-monotonous, also the change is smaller than the width of the distributions. Is this really significant? The same question for the minor change of the Flicker noise exponent from 1.5 to 1.6. I doubt that when repeating the experiments on another day, with another tip one would be able to reproduce the peak positions of the conductance histograms or the noise characteristics with the claimed precision.

Bottom of page 12: "Moreover, this separation of photoresponsive components from the electron transport route establishes a foundation for exploring multi-channel conductance mechanisms in single-molecule junctions." This is simply blabla.

Nothing has been separated here, there is no multichannel conductance mechanism.

About capacitive vs resistive transport: These are all DC transport measurements Nothing can be concluded about capacitive effects here. It is just lower and higher resistances. That's all.

The whole claim about transistors implementations, BJT vs FET is meaningless and misleading. In a transistor the current between two electrodes is controlled by a current or a field or a voltage applied to a third electrode. This is not at all what is done here. The molecules are investigated in different states which are to some extent controlled by irradiation. That's all. The molecules are irradiated as a whole. There is no specific action to the "gate electrode". Furthermore, the irradiation and relaxation times are in the range of 10s or 100s of seconds. The same function can be obtained by using "usual" photochromic molecules that can be toggled between two states.

There are many many language issues, ranging from simple grammar mistakes (singular/plural issues), via wrong technical terms (e.g. transportation instead of transport) to sentences that simply make no sense. One example out of many: "A 635 nm laser was placed outside of the main testing setup to minimize the vibration and noise, which was delivered to the STM-BJ cell through a long optical fiber." what was delivered through the optical fiber? The noise? A send example: "In contrast, the electron transport through the π -bridge pathway on SDAS relies on a distinct main-chain modulation mechanism (Fig. 2b)" How can the transport rely on a modulation mechanism? Probably the authors mean that the change of the transport relies on a modulation mechanism.

I did not read the 106 pages long supplementary information in all details. However, I saw also here some major flaws that affect the whole analysis. An example: The calculation of the resistances out of the conductance values on pages S32ff seem to be incorrect. A conductance of $G = 10^{(-3.9)} G_0$ corresponds to about 100 MOhm and not 102 kOhm. All values seem to be totally off. $1 G_0$ is ~12.9 kOhm, $0.1 G_0$ (i.e. $\log G/G_0 = -1$) is 129 kOhm. Hence 102 kOhm is in the order of $\log G/G_0 = -0.8$ and not $\log G/G_0 = -3.9$, as is claimed there.

The above list of criticisms is by far incomplete. This manuscript has to be rejected. I do not see any chance for a revision to result in an acceptable version, since it contains many unsupported big claims and misconceptions. It is lacking a clear description of the experiment, in particular for the single-anchored and the triple anchored molecules.

Version 2:

Reviewer comments:

Reviewer #1

(Remarks to the Author)

The authors have done a good job revising the manuscript. I am, however, wondering about one thing: The HR-MS (ESI) data look fine for all compounds, while none of the HR-MS (MALDI-TOF) data fit the calculated m/z. What is the reason for this?

Reviewer #2

(Remarks to the Author)

The authors have carefully addressed all the review comments and made sufficient revisions to the manuscript. I am pleased to recommend the manuscript for acceptance.

Reviewer #3

(Remarks to the Author)

2nd report about ms Fanxi Sun et al.

The authors have thoroughly revised their manuscript and have removed the most problematic parts of their material (e.g. the simulations which were totally wrong and clarified several unclear aspects.

Still, I find the manuscript highly intriguing, misleading and unpublishable.

The general impression remains that the work was and is premature and that the whole story is not sufficiently thought through. I am deeply convinced that the scientific work should be done by the authors and not by the reviewers of a manuscript. Throwing tons of data onto the readership and the reviewers (in the very lengthy rebuttal letter and revised supplementary information) and letting them fish out the important parts is simply impolite and does not convince me at all. Rather, it indicates that the authors do not really know which parts are really important for conveying their message.

Below I will give a selection of major concerns that have not been settled.

About original comment 2

The author's reply fixes the strictly wrong statements in the original manuscript but does not address the main problem: How can molecules be useful that require steady irradiation?

About original comment 3

In their response the authors agree that they explore not more than two conductance pathways. Still, in their revised material they insist in claiming multiple pathways and multiple anchoring. The latter could be fixed by writing "multiple anchoring possibilities", what would be not new though.

About original comment 4

The answer is not to the point. Fig. 1 a shows three electrodes to the molecule. This suggests that the same molecule would be contacted by three electrodes simultaneously This is simply wrong. The figure is more than misleading. It is pretending another experiment than what is done. Furthermore, throughout the whole manuscript it is not clearly stated what they are doing: I understood the experiment as follows: They have a three-legged molecule and believe that it is contacted to the bottom electrode by one of these legs and with the tip (that may have several protrusions) two of these legs. Is this a correct interpretation?

The comparison with the piezoelectric drive in Fig. 1b is additionally misleading, but probably not wrong.

The major problem of the misleading figure cannot be healed by rewriting the figure caption. Furthermore, the figure caption is contradicting what is shown and what the authors replied in an earlier comment: If all measurements were done in the dark, what's about 635 nm irradiation?

Additional questions. What is the meaning of the blueish or transparent bubbles?

About original comment 5

Also in their revised material the authors claim that the HOMO would be delocalized over the whole length (see e.g. response to reviewer 2, comment 10). This is wrong (or at least not supported by evidence). There is no doubt that the HOMO might be delocalized in some molecules (the long list of examples provided by the authors). But this is not to the point. For their argument it would be important to show that in this particular case studied here the HOMO is indeed delocalized over the whole length)

About original manuscript 6

The answer does not reply to the question., Probably my original comment was not clear. I meant: "For the single-anchored molecules: How do the authors know where the top tip contacts the molecule?"

About original comment 7

“About “V shape” dependence of the conductance on the bias voltage: There are only three data points: 100 mV, 300 mV, 500 mV. It is exaggerated to talk about V shape. It seems to be non-monotonous, also the change is smaller than the width of the distributions. Is this really significant? The same question for the minor change of the Flicker noise exponent from 1.5 to 1.6. I doubt that when repeating the experiments on another day, with another tip one would be able to reproduce the peak positions of the conductance histograms or the noise characteristics with the claimed precision.

Response: Thanks to the reviewer for the kind comments.

(1) Bias-dependent conductance. We appreciate the reviewer’s concern. Most single-molecule conductance studies report results measured at only one bias voltage, as summarized in the review on photoswitchable junctions (Chem. Commun. 2023, 59, 12685). Apart from a few specialized investigations focusing on bias-dependent effects (J. Am. Chem. Soc. 2023, 145, 21679-21686), systematic multi-bias analyses remain uncommon. Considering this, we have removed the term “V shape” and now describe the bias dependence simply as an objective experimental observation. For clarity and conciseness, the main text now focuses on the data collected at 300 mV. “

New comment: Other authors do not claim a V dependence of the conductance at all. Hence, these author do not need to provide evidence. Still, I appreciate the authors removed this overinterpretation of the data.

(2) Reproducibility. To ensure that the observed conductance features are reliable, each molecule was tested under five independent and orthogonal conditions, including three bias voltages under irradiation and in the dark as well as three independent datasets at 300 mV (in addition to the initial 300 mV measurement, we performed two additional light-off/light-on measurement cycles, resulting in three complete datasets at 300 mV). The resulting conductance histograms reproduce highly consistent conductance features across all conditions. (Fig. R8-R10 and Supplementary Figs. 32-45). The orthogonal repetition of the measurements substantially reinforces the reproducibility of the observed features.

New comment: This answer is not to the point. I did not ask about the reproducibility of the conductance measurements. (I raised this aspect of reproducibility only for the Flicker noise measurements My question about the statistical significance remains unanswered. If the distributions are wider than the separation of their main values, some analysis of the statistical significance has to be provided.

About original comment 8

I appreciate the revision. Still, the last sentence is overselling. The term “multiple” suggests “two or more” and is therefore overselling. The authors study the possibility of exactly two pathways and not more.

About original comment 9

I appreciate the revision which reduces the amount of misleading and overselling statements considerably, Still, there is simply NO analogy to any transistor, as the authors agree on in their response. Instead, in the revised abstract they suggest this analogy again. So, this issue is not solved.

Version 3:

Reviewer comments:

Reviewer #1

(Remarks to the Author)

The authors have removed the MALDI MS results that seemed to be wrongly calibrated. I therefore recommend acceptance of the manuscript.

Reviewer #2

(Remarks to the Author)

Reviewer #3

(Remarks to the Author)

3rd report about ms Fanxi Sun et al.

The authors have again revised their manuscript, and slowly things become clearer. In particular, several suggestive and misleading wordings and illustrations have been improved and exaggerated claims have been moderated.

1. For instance, Fig. 1a now shows that they do study single-molecule junctions in two-contact geometry (instead of three as in the original Fig. 1a). Interestingly, by revising Fig. 1a the authors have also changed the initially assumed contact

geometry by inverting the orientation of the molecule in the junction. I agree that the current orientation fits better to the interpretation of the stretching curves (short junctions corresponding to donor pathway, long junctions to the pi bridge configuration).

2. Unfortunately, in some figure (right-most illustration of Fig. 1d) they still show three electrodes contacting the molecule. According to their response to my previous comments, they agree that the molecule is only coupled to two electrodes simultaneously. Still, this is not explicitly stated in the manuscript. Furthermore, they authors still use the term “multi-anchored” in Fig. 1d and “triple-anchored” at several places in the text, although there is no triple-anchored junction. The molecule has three anchoring sites, yes, but only two of them are used simultaneously. The switching between the donor and the pi-bridge configuration is done by stretching, i.e., one contact to one anchoring site breaks and a contact to another one is formed. This is evidenced by the two-plateau shape of the stretching curves and the data is consistent with this interpretation. The two conduction pathways share one leg that remains stable and one that is swapped. Also this simple fact should be written down explicitly.

3. The photogating mechanism is a change of the molecular conformation that goes along with a change of the electronic structure. Thereby the conductance of the long pi bridge pathway is modified by more than a decade and the one of the short donor is only slightly modified (by around 0.2 decades and hence less than the width of the corresponding histogram peak). This is all nicely evidenced by the photochemical and transport investigations. The fact that both pathways are affected is not surprising at all, but still a nice functionality and has to the best of my knowledge not been reported before. Still, selling it as “simultaneous photogating of two pathways” is overselling: One pathway is affected, the other one is mainly unaffected.

4. Wordings like “... their modular structure that allows for triple-anchor configuration” and the comparison to transistors suggests that there would be three contacts simultaneously to one molecule. This is not the case and gives a completely wrong interpretation of the functionality. The worst is Fig. 6c, which draws again three terminals to the molecule. The described experimental evidence has nothing to do with transistor action at all. A transistor is a three-terminal device. The molecule studied here (the “triple anchored” one) is a device that has two different possibilities for a two-terminal configuration (the donor pathway and the pi bridge pathway) by keeping one anchoring group the same and changing the other one. The pi bridge pathway in both conformations (linear and cyclic) is very resistive, the donor pathway has a lower resistance in both conformations (also not surprising since the pi bridge is much longer). The resistance change to the pi bridge pathway upon photoisomerization is more pronounced than the one to the donor pathway (also not surprising since the conformational change takes place in the pi bridge part). The general situation remains the same: Highly resistive pi bridge pathway and low-resistive donor pathway. The similarity of both situations is also supported by the noise measurements: The difference between the linear and the cyclic case are not significant. There is absolutely no evidence to claim a transistor action at all and even less to claim a switching to another type of transistor. The comparison with transistors must be removed from the article, since it is scientific nonsense. Btw, assigning specific resistances to the individual arms is also highly suggestive and indicating the resistances with four digits is simply not justified by the data. And another remark: I am not sure if the figure caption of Fig. 6 a and b really fit to what is shown. I do not see red and blue arrows and I also do not know what a negative and a positive transport channel would be.

5. Furthermore, claiming “The π -bridge hybrid (through-bond/through-space mixed) transport in the linear state shifts to cyclopentenone-driven through-space transport in the cyclic state under light irradiation.”) is also wrong. Both have mixed character as shown by the noise data.

6. About original comment 6: Within the lengthy response, the only sentence which is to the point is: “For single-anchored molecules, in principle, a molecular junction could be formed between one thiomethyl anchor and a secondary, weaker interaction involving C, N, or O.”. Or in other words: The contact geometry is not really known. Although this is a minor point, it should be stated somewhere.

Summarizing, the authors admitted that some of their original claims were wrong or exaggerated and turned them down. Still, after two rounds of revision major flaws remain (transistor comparison, claim of switching of transport character). I repeat and extend one of my remarks of my previous (2nd) report: “The general impression remains that the work was and is premature and that the whole story is not sufficiently thought through. I am deeply convinced that the scientific work should be done by the authors PRIOR TO SUBMISSION and not by the reviewers of a manuscript.”

If the transistor part is removed from the manuscript and the other incorrect statements are corrected, the manuscript, I do not object against publication. If the remaining parts are substantial enough to warrant publication in Nature Communications is an editorial decision.

Reviewer #1

This is a highly interesting paper on how to modulate molecular conductance of a triple-anchored molecular photoswitch in a junction. The conductance measurements are very systematic and seem to be done excellently with proper reference measurements, but I should emphasize that I am not a physicist and not able to judge whether any additional test measurements are necessary for the conclusions. I can only say that I find the results very exciting and clearly presented, and I think this work should have a high impact.

Response: We sincerely thank the reviewer for the positive and encouraging comments. Following the reviewers' suggestions, we have carried out the required additional experiments and revised the manuscript, with all changes marked in red.

Comment 1

I only have one issue that I think needs to be addressed before publication. All new compounds must be fully characterized by ^1H -NMR, ^{13}C -NMR and HRMS or EA. In particular, for the final MeS-functionalized Stenhouse adducts, high-resolution mass spectra or elemental analysis should be provided to meet the highest standards of the synthesis community.

Response: Thanks to the reviewer for the kind comments. We appreciate the reviewer's careful attention to the characterization completeness. In the revised submission, newly synthesized intermediate and the five DASA derivatives have been fully characterized by ^1H NMR and ^{13}C NMR. Furthermore, for all six new molecules, high-resolution mass spectra (MALDI-TOF and ESI) have been newly acquired and incorporated in the Supplementary Information (**Supplementary Figs. 2, 4-8, 55, 56, and 72-83**).

Comment 2

A minor comment, the authors list molecular photoswitches that have been studied in junctions. Also dihydroazulene could be added to this list (e.g., *Nat. Commun.* 2017, 8, 15436).

Response: Thanks to the reviewer for the kind comments. The dihydroazulene photoswitch, which has been investigated using MCBJ techniques (*Nat. Commun.* **2017**, 8, 15436), has now been added to the Introduction (second paragraph) and cited as reference 27 in the revised manuscript.

Reviewer #2

In this article, the authors demonstrated light-controlled intramolecular electron transport through multiple conductive pathways in single DASA molecules using the STM-BJ technique. The results highlight the unique potential of DASAs as molecular transistor components, owing to their intrinsic responsiveness to near-infrared light and modular structure that enables multi-anchor configurations. I recommend this work for publication after revision. Several comments and suggestions are provided below:

Response: We greatly appreciate the reviewer's encouraging assessment of our study and the recognition of the advantages of DASA molecular systems in multi-pathway, light-controlled conductance. We thank the reviewer for recommending our work for publication after revision. Following the reviewers' suggestions, we have carried out the required additional experiments and revised the manuscript, with all changes marked in red.

Comment 1

Error bars should be added to Figure 3g to improve data reliability.

Response: Thanks to the reviewer for the kind comments. We performed additional light-on/light-off cycling measurements (two full cycles) at 300 mV (**Supplementary Fig. 35**) in addition to the original dataset. The average conductance values together with the corresponding standard deviations (**Table R1**) have now been included as error bars in the revised **Fig. R1**.

Table R1. Repeated light-on/light-off switching measurements of SSDA at 300 mV.

Measurement	linear SSDA (log G/G_0)	cyclic SSDA (log G/G_0)
1	-3.91	-3.71
2	-3.92	-3.60
3	-3.88	-3.64
Average	-3.90	-3.65
Standard deviations	0.02082	0.05568

Fig. R1 | Summary of conductance for *linear* and *cyclic* SSDA. Conductance for *linear* and *cyclic* SSDA at bias voltages of 100 mV, 300 mV and 500 mV.

Comment 2

The authors achieved photoconversion from linear to cyclic molecular structures upon light irradiation, with reversion to the linear form after the light is turned off. Is this photoresponse fully reversible? If an irreversible component exists, what is its approximate proportion? Furthermore, can the reversible switching between linear and cyclic forms be sustained over multiple cycles? I suggest a repeatedly isomerization experiment should be done. Does repeated photochemical switching lead to molecular degradation or loss of functionality?

Response: Thanks to the reviewer for the kind comments. To evaluate the reversibility and fatigue resistance of the photoisomerization process, we performed repeated *linear-to-cyclic* and *cyclic-to-linear* switching experiments for **SSDA**, **SDAS** and **SSDAS** using both real-time UV-vis spectroscopy (**Figs. R2-R7**) and STM-BJ measurements (**Figs. R8-R10**).

For real-time UV-vis spectroscopy, in all three molecules, the main absorption band shows only a slight decrease in intensity after ≥ 6 switching cycles, indicating minimal photodegradation and confirming that the *linear* state is largely recovered after sufficient thermal relaxation in the dark (**Figs. R2-R4**). Moreover, no new absorption features appear across the entire monitored spectral range during repeated photoisomerization (**Figs. R5-R7**), further demonstrating the chemical integrity of each DASA derivative throughout cycling.

Corresponding STM-BJ measurements under repeated light-on/light-off conditions (**Figs. R8-R10**) were also performed at 300 mV bias. For each DASA derivative, both the *linear* and *cyclic* conductance states persist across switching cycles, and the characteristic conductance signatures remain well distinguishable (**Figs. R8-R10**). Importantly, no systematic drift or loss of conductance features was observed, demonstrating that repeated photoisomerization does not cause measurable molecular degradation.

Together, these UV-vis and STM-BJ results confirm that the photoresponse of **SSDA**, **SDAS**, and **SSDAS** is highly reversible, with only minor irreversible components after several switching cycles (**Figs. R2-R4**).

Fig. R2 | Reversible photoswitching cycles of SSDA. Fatigue resistance of SSDA under repeated *linear-to-cyclic* photoisomerization (635 nm laser, 1 mW cm⁻²) and *cyclic-to-linear* thermal relaxation (19 °C, dark). UV-vis absorbance at 656 nm was monitored over multiple switching cycles.

Fig. R3 | Reversible photoswitching cycles of SDAS. Fatigue resistance of SDAS under repeated *linear-to-cyclic* photoisomerization (635 nm laser, 1 mW cm⁻²) and *cyclic-to-linear* thermal relaxation (19 °C, dark). UV-vis absorbance at 636 nm was monitored over multiple switching cycles.

Fig. R4 | Reversible photoswitching cycles of SSDAS. Fatigue resistance of SSDAS under repeated *linear-to-cyclic* photoisomerization (635 nm laser, 1 mW cm⁻²) and *cyclic-to-linear* thermal relaxation (19 °C, dark). UV-vis absorbance at 657 nm was monitored over multiple switching cycles.

Fig. R5 | Time-dependent UV-vis spectra of SSDA during repeated photoswitching cycles.

Fig. R6 | Time-dependent UV-vis spectra of SDAS during repeated photoswitching cycles.

Fig. R7 | Time-dependent UV-vis spectra of SSDAS during repeated photoswitching cycles.

Fig. R8 | Reversible conductance switching of SSDA at 300 mV bias. a Chemical structures and reversible *linear-cyclic* isomerization of SSDA. **b** Two-dimensional conductance-distance histograms of *linear* SSDA (purple) and *cyclic* SSDA (grey) junctions measured at a bias voltage of 300 mV; the color scale represents the number of counts. Insets show the relative stretching distance distributions for *linear* SSDA and *cyclic* SSDA. **c** One-dimensional logarithmic conductance histograms of *linear* (purple) and *cyclic* (grey) SSDA recorded under “light off” and “light on” conditions at 300 mV bias.

Fig. R9 | Reversible conductance switching of SDAS at 300 mV bias. **a** Chemical structures and reversible *linear-cyclic* isomerization of SDAS. **b** Two-dimensional conductance-distance histograms of *linear* SDAS (blue) and *cyclic* SDAS (gray) junctions measured at a bias voltage of 300 mV; the color scale represents the number of counts. Insets show the relative stretching distance distributions for *linear* SDAS and *cyclic* SDAS. **c** One-dimensional logarithmic conductance histograms of *linear* (blue) and *cyclic* (gray) SDAS recorded under “light off” and “light on” conditions at 300 mV bias.

Fig. R10 | Reversible conductance switching of SSDAS at 300 mV bias. **a** Chemical structures and reversible *linear-cyclic* isomerization of SSDAS. **b** Two-dimensional conductance-distance histograms of *linear* SSDAS (yellow) and *cyclic* SSDAS (gray) junctions measured at a bias voltage of 300 mV; the color scale represents the number of counts. Insets show the relative stretching distance distributions for *linear* SSDAS and *cyclic* SSDAS. **c** One-dimensional logarithmic conductance histograms of *linear* (yellow) and *cyclic* (gray) SSDAS recorded under “light off” and “light on” conditions at 300 mV bias.

Comment 3

The simulation section in Figure 6 should clearly state the input parameters and the rationale behind the equivalent resistance calculations. It is recommended to include these details in the main text or the Supplementary Information.

Response: Thanks to the reviewer for the kind comments. After reconsidering the simulation in Fig. 6, we decided to remove it because the simulated data were not essential to the conclusions of the work and could introduce unnecessary model-dependent uncertainty. The key point we intended to illustrate is that the *linear* triene π -bridge exhibits more “resistive-like” through-bond transport, whereas the *cyclic* cyclopentenone unit favors more “capacitive-like” through-space transport. This qualitative interpretation is fully supported by the experimental STM-BJ measurements and the DFT electronic-structure analysis. For completeness, the details of the original equivalent-resistance conversion are provided in Section 6 of the Supplementary Information. The corresponding discussion in the main text has also been revised or removed as appropriate, and all modifications are highlighted in red.

Comment 4

Physical quantities in the figures and main text should be italicized.

Response: Thanks for your kind comments. We have carefully double-checked the main text, the Supplementary Information, and all figures, and have corrected all physical-quantity symbols to italic formatting. This includes E (molecular energy), G , G_0 , E_F (Fermi energy), k (rate constant), Dz (pulling distance), I (current), V (voltage), ε (molar absorptivity) and other relevant physical variables. All such symbols now follow the required typographic convention.

Comment 5

The abstract is somewhat lengthy and could be condensed to more effectively emphasize the innovative aspects and key findings.

Response: Thanks to the reviewer for the kind comments. Following the recommendation, we have substantially revised and condensed the abstract to more clearly highlight the key findings and the innovative aspects of our work. The revised abstract (188 words) is provided below:

“Manipulating intramolecular electron transportation can fundamentally modulate the optical property, electromagnetic behavior and chemical reactivity of molecules. Achieving simultaneous control over multiple transport pathways within a single molecule, however, remains a significant challenge. Herein, we report light-gated modulation of two distinct conductive pathways in single donor-acceptor Stenhouse adduct (DASA) molecules using the scanning tunneling microscopy break-junction (STM-BJ) technique. The donor and π -bridge pathways were separately controlled by fabricating double-anchored DASAs with thiomethylic groups substitution: (1) the donor pathway is manipulated on side-chain mechanism, where the linear-to-cyclic isomerization induces electronic redistribution and increases the conductivity;

(2) the π -bridge pathway is manipulated on main-chain mechanism, and the deformation of π -conjugation decreases the conductivity. By synthesizing triple-anchored DASAs, these two conductive pathways were integrated into single-molecule junctions and could be simultaneously modulated under 635 nm red light irradiation and dark relaxation. This photoinduced shift from π -bridge hybrid (through-bond/through-space mixed) transport in the linear state to cyclopentenone-driven through-space transport in the cyclic state renders the two isomers structurally analogous to bipolar junction transistor (BJT)-like and metal-oxide-semiconductor (MOS)-like configurations. These results highlight DASAs' potential in understanding molecular electronics and developing photoresponsive molecular-scale devices.”

Comment 6

Although schematics are provided comparing “side-chain” and “main-chain” modulation mechanisms for SSDA and SDAS, adding a graphical abstract would help visually summarize the light-controlled dual-channel switching mechanism in SSDAS.

Response: Thanks to the reviewer for the kind comments. The light-controlled dual-channel switching mechanism of SSDAS is already illustrated in the revised Fig. 5a, b, where both conductive pathways in the *linear* and *cyclic* states are resolved through the corresponding 2D conductance-distance maps. These plots directly visualize the coexistence and modulation of the two transport channels, which we believe serves the purpose of a graphical abstract within the context of this figure.

Fig. R11 | The revised Fig. 5.

Comment 7

The “BJT-like” and “MOS-like” analogies are creative but should be used cautiously and accompanied by detailed explanations to avoid oversimplifying the differences between molecular-scale devices and macroscopic electronic components.

Response: Thanks to the reviewer for the kind comments. In response, we have removed the transistor-related analogies that relied on simulated or quantitative interpretations, and we now retain only a brief structural analogy to highlight the qualitative differences between the *linear* and *cyclic* configurations (**Fig. R12**). All overstated or potentially misleading comparisons to BJT- and MOS-type behavior based on the previous simulation have been deleted, and the corresponding sections in the main text have been revised accordingly. The modifications are marked in red in the revised manuscript.

Fig. R12 | The revised Fig. 6.

Comment 8

The conclusion could be strengthened by further emphasizing the potential implications of this work for the development of multi-state logic devices or optoelectronic intelligent materials.

Response: Thanks to the reviewer for the kind comments. Following the recommendation, we have revised the conclusion to clearly emphasize the broader implications of light-controlled multi-pathway modulation, particularly for the development of multi-state molecular logic functions and optoelectronic intelligent materials. The revised conclusion is more concisely highlighting the conceptual advances and potential applications. All modifications have been marked in red in the revised manuscript. The revised conclusion is provided below:

“In summary, this study offers opportunity to precisely control the molecular electronic properties by photogating multiple conductive pathways in single-molecule junctions, where the electron transportation through the photoresponsive molecules could be switched and modulated by light. A series of thiomethylic-groups-substituted DASAs were synthesized for the construction of single-molecule junctions, including the single (DAS and SDA), double (SSDA and SDAS) and triple-anchored (SSDAS) molecules. Firstly, the electron transportation either through the donor pathway and π -bridge pathway is separately investigated on the double-anchored DASAs: (1) the donor pathway is achieved on SSDA, where the electrons traverse the diphenylaminic donor, the linear-cyclic isomerization varies the conjugated structure on the side chain and regulates the conductance on the main chain within 0.20 orders of magnitude; (2) the π -bridge pathway is achieved on SDAS, where the electrons travers the entire molecule (i.e. donor, π -bridge and acceptor), the formation/deformation of the π -bridge on the main chain switches the conjugated structure between hybrid (through-bond/through-space mixed) transportation in linear and through-space transportation in cyclic, which generates a conductance variation within 0.62 orders of magnitude. The donor and π -bridge pathways were then integrated into one single-molecule junction by constructing triple-anchored SSDAS, which could be simultaneously modulated under controlling of 635 nm red light irradiation, demonstrating the possibility of photogated multiple conductive pathways in single-molecule junctions. As the electron transportation in SSDAS retains the characteristics for both the donor pathway (side chain modulation) and π -bridge pathway (main chain modulation), a structurally transistor-like behavior was demonstrated. When SSDAS is in its linear form, the transistor exhibits BJT-like structure where the π -bridge pathway (conjugated π -bridge) with a relatively lower resistance of 2297 M Ω exhibits “resistive-regime” transport; after the red-light-induced linear-to-cyclic isomerization, the resistance of the π -bridge pathway (cyclopentenone) sharply increases to 30220 M Ω , which exhibits “capacitive-regime” tunneling, switching the transistor into a MOS-like structure.

These results establish a molecular platform in which multiple electron-transport modes can be selectively addressed by light, offering opportunities for multi-level conductance states and optically gated logic behaviors at the molecular scale. Such capabilities provide a conceptual foundation for the design of adaptive and stimuli-responsive molecular electronic elements. However, challenges remain in achieving precise control over electrode-molecular junction geometries and scalable integration into device platforms. Future work combining in situ spectroscopic characterization, advanced simulation and multi-functional junction architectures will be critical to fully harness the potential of photoresponsive molecules in nanoelectronics.”

Comment 9

The x-axis of the inset figures of typical individual conductance-distance traces in Figure 3d and 3e should be added.

Response: Thanks to the reviewer for the kind comments. In response, we have added the x-axis to all individual conductance-distance traces shown alongside the 2D conductance histograms in the main figures, including **Figs. 3b, c, 4d, c, 5a, b (Fig. R13)**, as well as **Supplementary Figs. 35b, 38b, and 45b**. These panels now clearly display the pulling distance on the x-axis for the representative single-trace examples.

For **Supplementary Fig. 46**, we intentionally present a collection of 12 individual conductance-distance traces without the x-axis. This figure is designed specifically to illustrate the diversity of anchoring geometries in multi-anchored molecules, and the inclusion of many traces in a compact layout would make x-axis labels difficult to read without improving interpretability. Because the purpose of this figure is to show variability across anchoring configurations (which is already supplied in the x-axis-labeled figures listed above), we have kept **Supplementary Fig. 46** in its original simplified format.

Fig. R13 | Revised 2D conductance-distance maps and representative individual conductance-distance traces of SDAS, SSDA, and SSDAS. Combined plots showing the 2D conductance-distance histograms and its corresponding individual conductance-distance traces for SDAS (a, b), SSDA (c, d), and SSDAS (e, f), with unified Dz axes added for clarity.

Comment 10

It is better to have some discussions about why the conductance of cyclic-SSDA/SDAS is larger than that of linear-SSDA/SDAS. Since the length of cyclic type molecule is even longer than that of linear type, and the main tunneling route of those molecules are the same of benzene-N-benzene, what's the determining point of Nitrogen atom to affecting the conductance? Electronegativity or anything else?

Response: Thanks to the reviewer for the kind comments. The conductance difference between the *linear* and *cyclic* isomers of SSDA/SDAS can be better understood by considering how photoisomerization reshapes the frontier orbital distributions along the dominant transport pathways, as supported by STM-BJ measurements, molecular lengths and theoretical transmission analyses (**Fig. R14a-c**).

- (1) **SSDA: a fixed donor-pathway dominated transport.** For SSDA, both *linear* and *cyclic* isomers share the same primary tunneling route, **the donor-side phenyl-N-phenyl pathway**. Photoisomerization only induces a slight geometric contraction ($10.26 \text{ \AA} \rightarrow 10.18 \text{ \AA}$), indicating that the backbone topology remains essentially unchanged (**Fig. R14a**). However, the HOMO distribution exhibits a notable shift: in the *cyclic* isomer, the orbital density becomes more concentrated on the donor segment (**Fig. R14a**). This redistribution enhances the effective electronic coupling along the phenyl-N-phenyl channel, leading to the experimentally observed modest conductance increase. DFT transmission spectra (**Fig. R14b**) show that the *cyclic* SSDA maintains a stronger resonance tail near the Fermi level, consistent with this behavior.
- (2) **SDAS: an adjustable main-chain pathway with a different mechanism.** In SDAS, charge transport occurs through an **extended main-chain route** that involves **donor** \rightarrow **conjugated π -bridge/cyclopentenone** \rightarrow **acceptor anchoring group**. HOMO maps demonstrate substantial density over the donor and π -bridge, but much weaker delocalization through the pyrazole-N toward the acceptor anchoring site (**Fig. R14a**), which makes SDAS intrinsically lower in conductance. Upon ring closing, the π -bridge is converted to a cyclopentenone, which shortens the molecular length ($15.18 \text{ \AA} \rightarrow 11.86 \text{ \AA}$) but simultaneously diminishes overall conjugation. This structural change leads to a further decrease in conductance, as reflected by a deeper transmission suppression near the Fermi level (**Fig. R14c**).

Although the molecular lengths of the two isomers differ, the decisive factor lies in **how isomerization modulates orbital overlap and delocalization along the dominant transport pathway**, rather than a single atomic property. Specifically, **SSDA: *cyclic* isomer enhances orbital concentration on the fixed donor pathway \rightarrow slightly higher conductance. SDAS: *cyclic* isomer reduces π -conjugation along the adjustable main-chain pathway \rightarrow lower conductance.** The discussion addressing the above question has been added to the main text as

follows:

“A comparison with SSDA further clarifies how photoisomerization differs between these two systems. For SSDA, both the isomers preserve the same donor-side benzene-N-benzene transport pathway, and HOMO maps (Fig. 2a and 2b) show that the cyclic form exhibits a more concentrated orbital distribution on this fixed donor segment compared to the linear form. This redistribution slightly strengthens the electronic coupling while keeping the backbone topology essentially unchanged, which accounts for the modest conductance enhancement. In contrast, photoisomerization directly reshapes the π -bridge of SDAS. For the linear isomer, the HOMO is delocalized across the donor and the extended π -system, supporting a continuous channel. After isomerized to the cyclic form, the π -bridge is disrupted and the HOMO becomes more localized, leading to reduced conjugation and lowered conductance.”

Fig. R14 | Pathway-dependent conductance modulation in SSDA and SDAS. **a**, Single-molecule conductance ($\log(G/G_0)$) distributions of four representative molecules (*linear SSDA*, *cyclic SSDA*, *linear SDAS* and *cyclic SDAS*), highlighting distinct charge-transport pathways governed by structural topology. **b**, **c** The theoretical calculated transmission curves of (b) *linear SSDA* (purple) and *cyclic SSDA* (gray) (c) *linear SDAS* (blue) and *cyclic SDAS* (gray).

Reviewer #3

Comment 1

Referee report about ms. Photogated Multiple Conductive Pathways of Donor-Acceptor Stenhouse Adducts in Single-Molecule Junctions by Fanxi Sun et al. The manuscript reports the fabrication and characterization of phototunable donor acceptor Stenhouse-based molecules as well as STM breakjunction measurements and accompanying DFT calculations. The authors claim that some of these molecules may act as equivalents to transistors. The manuscript contains major flaws and cannot be published. The problems start already in the introduction where the authors claim that the plasma resonances of typical metals used for contacting single molecules would be in the UV range. This is simply wrong as a simple literature survey shows. They also claim that other phototunable molecules would be addressable in the UV range only. This is wrong as well. The most important classes of photochromic molecules with two distinct states can be activated also in the visible range (for one of the switching directions). They mention several drawbacks of activation in the UV and miss to mention the most important one of them: photodegradation.

Response: Thanks to the reviewer for the kind comments. The original wording may indeed lead to misunderstanding, and we thank the reviewer for pointing this out. We have revised the introduction carefully to remove any potential overgeneralization and to present the statements more accurately.

(1) **Regarding the description of plasmonic resonances**, our original intention was not to claim that the plasmonic resonances of **all metals** used in molecular junctions **are strictly** located in the UV region. Rather, we intended to highlight that **for noble metals such as Au and Ag, the localized surface plasmon resonances (LSPRs) typically lie at the UV-visible boundary (around 350-520 nm)**, while the 635 nm laser source largely avoids this spectral overlap.

(2) **Regarding the activation wavelength of photochromic molecules**, we also agree that several classes of photochromic molecules such as azobenzene, spiropyran, and diarylethene can be switched in the visible region for one of the isomerization directions. **Our statement referred to the fact that many conventional systems still require UV activation for at least one transition.** Therefore, although several reported photoswitches exhibit one-direction-isomerization under triggering by visible light, UV light is still necessary to control the reversible isomerization. In contrast, the photoactivation of DASAs is **entirely unrelated to UV excitation**, and therefore their operation inherently avoids UV-induced photobleaching. This clarification has been incorporated into the revised manuscript.

For one of the advantages of visible light control, our original sentence is:

“Secondly, most photoresponsive molecules employed in single-molecule studies rely on UV light ($\lambda < 380$ nm) to induce isomerization. Such short-wavelength irradiation leads to multiple complications, as it may trigger photoelectric effects, plasmonic resonance or localized heating on metallic electrodes (e.g., Au, Ag, Pt), thereby interfering with conductance measurements and reducing device stability.”

Has been revised as below:

*“Secondly, UV excitation ($\lambda < 380$ nm) is still required for **at least one switching direction** in many photoresponsive molecules used in single-molecule studies. Irradiation in this short-wavelength regime can introduce undesired effects such as **photodegradation**, photoelectric responses, plasmon-related perturbations or localized heating in metal nanogaps, which may interfere with conductance measurements and compromise device stability.”*

Comment 2

The Stenhouse molecules studied here are activated in the visible range, but apparently this state is not stable since they relax back in the dark within a timescale of a few hundred seconds. This is a major drawback for their usage. It also puts the results of the transport measurements in question, since they were performed in the dark and no time frame is indicated. So, it cannot be excluded that the molecules were already partly relaxed back. In the bottom of page 4 the authors mention near infrared activation, which is not investigated here.

Response: Thanks to the reviewer for the kind comments. We agree that the thermal relaxation of the *cyclic* state in the dark needs to be addressed clearly. In our STM-BJ experiments, the conductance of the *cyclic* state was measured **under continuous visible-light irradiation**, and the irradiation was **not interrupted** during data acquisition. Therefore, the relaxation pathway described by the reviewer (“back in the dark”) did not occur under our experimental conditions. Continuous irradiation ensures that the DASA molecules remain in the photoinduced *cyclic* form throughout the measurement period. This is further supported by the UV-Vis spectra (**Fig. 2c-e; Supplementary Fig. 15-19**) and by the **light-on/light-off evolution and repeatability tests (Supplementary Figs. 35, 38, and 45)**.

To avoid any possible misunderstanding, all expressions such as “before irradiation” and “after irradiation” have been revised to **“without irradiation”** and **“under irradiation”**, to explicitly reflect the experimental conditions during conductance measurements. The original sentence has been revised as follows:

*“2D conductance histograms reveal two distinct conductance bands in the range of $-4.5 \leq \log(G/G_0) \leq -2.5$ for both the **without-irradiation** and **under-irradiation** measurements (Fig. 3b and 3c).”*

“Summary of calculated (the position near zero energy) and measured conductance changes for SDAS, SSDA and SSDAS before and under red light irradiation.”

Regarding the comment on activation wavelengths, the earlier mention of “near-infrared light” has also been replaced with “**long-wavelength visible light (deep-red region)**”. The original sentence has been revised as follows:

*“These results highlight the unique potential of DASAs as molecular transistor components, stemming from their intrinsic responsiveness to **long-wavelength visible light (deep-red region)** and their modular structure that allows for multi-anchor configuration.”*

Comment 3

The authors repeatedly write about “multiple conductance pathways”. In fact they discuss 2 (in the molecule with three anchoring sites), not multiple, and even this claim is not supported by evidence, because it is not made clear how these three anchoring points can be realized in the junction.

Response: Thanks to the reviewer for the kind comments. In our manuscript, the term “multiple conductance pathways” refers to the experimentally observed coexistence of **two distinct transport routes** that can appear within a single STM-BJ pulling cycle. The evidence is presented in **Fig. R15**, where individual conductance traces for both the *linear* and *cyclic* SSDAS molecules display **two well-separated plateau regions**, high-*G* and low-*G*. Importantly, these two signatures can occur **sequentially within the same electrode retraction event** (**Fig. R15**), indicating that the junction can adopt different Au-molecule-Au bonding configurations during a single pulling cycle. Regarding the anchoring geometries are illustrated in **Fig. R15a, b** and are experimentally resolved through their characteristic plateau lengths and conductance levels.

Fig. R15 | Multiple anchoring-dependent conductance pathways revealed in individual STM-BJ pulling traces. a, b, Two-dimensional conductance-displacement histograms for (a) *linear* SSDAS and (b) *cyclic* SSDAS. **c,** Representative single-molecule conductance traces extracted from the donor, π -bridge, and dual-pathway regions, highlighting that a single pulling event may access multiple transport routes. **d-g,** Additional 2D conductance histograms and corresponding traces for (d, f) *linear* and (e, g) *cyclic* SSDAS.

Comment 4

Fig. 1a suggests three anchoring points and somehow also that the anchoring point might be changed by the piezo? This is all highly suggestive but not explained.

Response: Thanks to the reviewer for the kind comments. In our STM-BJ measurements, the “piezo” refers solely to the **piezoelectric actuator that controls the approach and retraction of the gold tip** relative to the gold substrate. It does **not** exert any mechanical manipulation on the molecule itself, nor does it switch anchoring points through a mechanical-electrical energy conversion mechanism. The term may indeed have caused unintended ambiguity. To avoid further confusion, we have revised the description of the STM-BJ setup in the Supplementary Information. The revised text now reads:

“All measurements were carried out in the dark at 298 K. During measurements, the gold tip was repeatedly brought into and out of contact with the substrate at 10 nm/s by a piezoelectric actuator.”

In addition, the schematic in the original **Fig. 1a** has been revised accordingly and is now provided as **Fig. R16** in the revised manuscript.

Fig. R16 | Single-molecule conductance measurement setup. Schematic illustration of the

in-situ laser-irradiated single-molecule conductance measurement setup, DASAs isomerize between *linear* and *cyclic* in a 1,2,4-trichlorobenzene (TCB) droplet under controlling of a 635 nm laser.

Comment 5

On page 8 the authors try to explain the change of the conductance of the main chain upon photocyclisation. They claim that in the linear state the HOMO would be delocalized while it would be localized in the cyclic form. In fact, Fig. 2b clearly shows that the HOMO is NOT extended over the whole linear chain. It is also interrupted almost in the middle of the molecule. In the cyclic form the HOMO is probably a bit more localized, **but the molecule is also much shorter.**

Response: Thanks to the reviewer for the kind comments. What we aimed to highlight is the **redistribution of the frontier orbitals arising from photoisomerization**, which reshapes both the electronic structure and the dominant transport pathways. This aspect has already been discussed in detail in our response to Reviewer #2, Comment 10, where we analyzed **how photoisomerization reshapes the frontier orbital distributions along the dominant transport pathways**, supported by STM-BJ measurements, molecular lengths and theoretical transmission analyses (**Fig. R14**).

Briefly, in the *linear* SDAS, the triene π -bridge contributes substantially to the HOMO (**Fig. R14a**, left), enabling a more conjugated main-chain transport route. Upon photocyclization, the conjugated triene is transformed into a cyclopentenone, resulting in a HOMO that becomes **more localized toward the donor segment** (**Fig. R14a**, right). This loss of π -conjugation is directly reflected in the transmission spectra (**Fig. R14c**), which exhibits a deeper suppression around the Fermi level for the *cyclic* isomer and thereby account for the reduced main-chain conductance. Conversely, the enhanced HOMO weight on the donor segment in the *cyclic* SSDA strengthens the fixed donor-pathway coupling, which explains the slightly higher conductance observed for the *cyclic* SSDA isomer.

We fully agree that molecular length plays an important role in determining conductivity. At the same time, and as widely recognized in molecular electronics, **length is one of several factors governing charge transport, together with orbital alignment, conjugation efficiency and electrode-molecule coupling.** As demonstrated in a representative study (*J. Am. Chem. Soc.* **2025**, 147, 24, 20310-20317), molecular wires can exhibit remarkably high conductance values approaching **1 G_0** over lengths exceeding **20 nm** under low bias, without any discernible decay with length.

Overall, the HOMO distributions shown are objective outcomes of the DFT calculations and

serve as an **essential basis** for understanding the subsequent analysis of transport pathways and conductance modulation. In addition, molecular length is only one of many parameters that influence single-molecule conductance. Below are several representative studies on **anti-Ohmic behavior, weak/negative decay**, and key factors governing molecular conductance:

- ✧ *Long-Range Charge Transport in Molecular Wires*
(*J. Am. Chem. Soc.* **2024**, *146*, 47, 32206-32221)
- ✧ *Zero-Bias Anti-Ohmic Behaviour in Diradicaloid Molecular Wires*
(*Angew. Chem. Int. Ed.* **2024**, *63*, e202410304)
- ✧ *Comprehensive Suppression of Single-Molecule Conductance Using Destructive σ -Interference*
(*Nature* **2018**, *558*, 7710, 415-419)
- ✧ *Long-Range Resonant Charge Transport through Open-Shell Donor-Acceptor Macromolecules*
(*J. Am. Chem. Soc.* **2025**, *147*, 20310-20317)
- ✧ *Near Length-Independent Conductance in Polymethine Molecular Wires*
(*Nano Lett.* **2018**, *18*, 6387-6391)
- ✧ *Origin of the Reversed Conductance Decay in Radical Cationic Molecular Wires: A Molecular Orbital Perspective*
(*J. Phys. Chem. C* **2024**, *128*, 21498-21507)

Comment 6

For the single-anchored molecules: How do the authors know where the top contacts the molecule?

Response: Thanks to the reviewer for the kind comments. For DASA, atoms such as **oxygen or nitrogen on the π -bridge or the acceptor segment** may interact with gold electrodes and form parasitic transport pathways. The single-anchored molecules DAS and SDA were specifically designed and measured as control experiments to determine whether **unintended anchoring** could contribute to the conductance features observed in SSDA and SDAS.

Under identical STM-BJ conditions, including three bias voltages (100, 300 and 500 mV) and both irradiation states, **neither DAS nor SDA exhibited conductance features similar to those of SSDA, SDAS and SSDAS (Supplementary Figs. 24-30)**. These results demonstrate that **non-thiomethyl anchoring does not generate the conductance signatures associated with donor or π -bridge pathways**. A detailed discussion of unconventional anchoring is provided in Section 6.2 of the SI.

Comment 7

About “V shape” dependence of the conductance on the bias voltage: There are only three data points: 100 mV, 300 mV, 500 mV. It is exaggerated to talk about V shape. It seems to be non-monotonous, also the change is smaller than the width of the distributions. Is this really significant? The same question for the minor change of the Flicker noise exponent from 1.5 to 1.6. I doubt that when repeating the experiments on another day, with another tip one would be able to reproduce the peak positions of the conductance histograms or the noise characteristics with the claimed precision.

Response: Thanks to the reviewer for the kind comments.

(1) Bias-dependent conductance. We appreciate the reviewer’s concern. Most single-molecule conductance studies report results measured at only **one bias voltage**, as summarized in the review on photoswitchable junctions (*Chem. Commun.* **2023**, 59, 12685). Apart from a few specialized investigations focusing on bias-dependent effects (*J. Am. Chem. Soc.* **2023**, 145, 21679-21686), systematic multi-bias analyses remain uncommon. Considering this, we have removed the term “V shape” and now describe the bias dependence simply as **an objective experimental observation**. For clarity and conciseness, the main text now focuses on the data collected at 300 mV.

(2) Reproducibility. To ensure that the observed conductance features are reliable, each molecule was tested under **five independent and orthogonal conditions**, including three bias voltages under irradiation and in the dark as well as three independent datasets at 300 mV (in addition to the initial 300 mV measurement, we performed two additional light-off/light-on measurement cycles, resulting in **three complete datasets at 300 mV**). The resulting conductance histograms reproduce highly consistent conductance features across all conditions. (**Fig. R8-R10 and Supplementary Figs. 32-45**). The orthogonal repetition of the measurements substantially **reinforces the reproducibility** of the observed features.

(3) Flicker noise exponent. Each PSD dataset contains more than **six thousand individual traces**, providing a robust statistical basis. The Flicker noise exponents for *linear* and *cyclic* SSDA, **1.51 and 1.61**, differ only slightly, which is reasonable because the fundamental **phenyl-N-phenyl transport pathway** remains unchanged. To avoid overinterpretation, we have revised the statement in the manuscript as follows:

“For both linear and cyclic SSDA, the noise power scales with conductance following a power-law dependence, with extracted exponents of $G^{1.51}$ and $G^{1.61}$ for the linear and cyclic isomers, respectively (Supplementary Fig. 34). According to previous reports⁵⁸, a PSD exponent close to 1.0 indicates a dominant through-bond transport mechanism, whereas values approaching 2.0 suggest increasing contributions from through-space tunneling. This interpretation is supported by transmission pathway calculation, which reveals that both isomers exhibit mixed through-

bond (blue arrows) and through-space (red arrows) transport channels (Supplementary Fig. 51). Therefore, the intermediate exponents (1.5-1.6) observed experimentally reflect a combined contribution of the two mechanisms.”

Comment 8

Bottom of page 12: “Moreover, this separation of photoresponsive components from the electron transport route establishes a foundation for exploring multi-channel conductance mechanisms in single-molecule junctions.” This is simply blabla. Nothing has been separated here, there is no multichannel conductance mechanism.

Response: Thanks to the reviewer for the kind comments. We appreciate the reviewer’s concern and have revised the wording to avoid any possible overstatement. The term “**separation**” means that within one molecule, under light irradiation, the high-conductance pathway increases from $10^{-3.90}G_0$ to $10^{-3.64}G_0$, while the low-conductance pathway decreases from $10^{-5.26}G_0$ to $10^{-6.37}G_0$ (Fig. R17). In one molecule, the two conductance pathways independently shift “**one higher and the other lower**” showing a clear separation. To avoid misleading, we have revised the statement in the manuscript as follows:

*“Moreover, **the opposite shifts** of the two conductance features upon photoisomerization highlight the distinct ways in which different parts of the molecule respond to light, offering a foundation for exploring multi-channel conductance mechanisms in single-molecule junctions.”*

Fig. R17 | One-dimensional logarithmic conductance histograms of *linear* SSDAS (upper, yellow) and *cyclic* SSDAS (lower, gray) at a bias voltage of 300 mV.

Comment 9

About capacitive vs resistive transport: These are all DC transport measurements. Nothing can be concluded about capacitive effects here. It is just lower and higher resistances. That's all. The whole claim about transistor implementations, BJT vs FET is meaningless and misleading. In a transistor the current between two electrodes is controlled by a current or a field or a voltage applied to a third electrode. This is not at all what is done here. The molecules are investigated in different states which are to some extent controlled by irradiation. That's all. The molecules are irradiated as a whole. There is no specific action to the "gate electrode". Furthermore, the irradiation and relaxation times are in the range of 10s or 100s of seconds. The same function can be obtained by using "usual" photochromic molecules that can be toggled between two states.

Response: Thanks to the reviewer for the kind comments. We appreciate the reviewer's critical perspective on this point. We fully agree that our STM-BJ measurements in this work are carried out under DC bias. We also note that STM-BJ techniques can, in principle, be extended to apply periodically modulated biases within a single hover event to investigate frequency-dependent transport, which we regard as an interesting direction for future studies, but this lies outside the scope of the present manuscript.

In the earlier version of the manuscript, the terms "resistive" and "capacitive" were used only as qualitative analogies to describe through-bond versus through-space dominated transport pathways. In response, **the simulation figure has been deleted**, and the remaining text has been carefully rewritten to avoid suggesting that a three-terminal transistor operation is realized here, and **we now retain only a brief structural analogy** to highlight the qualitative differences between the *linear* and *cyclic* configurations (**Fig. R12**). All overstated or potentially misleading comparisons to BJT- and MOS-type behavior based on the previous simulation have been deleted, and the corresponding sections in the main text have been revised accordingly. The modifications are marked in red in the revised manuscript.

Comment 10

There are many many language issues, ranging from simple grammar mistakes (singular/plural issues), via wrong technical terms (e.g. transportation instead of transport) to sentences that simply make no sense. One example out of many: "A 635 nm laser was placed outside of the main testing setup to minimize the vibration and noise, which was delivered to the STM-BJ cell through a long optical fiber." what was delivered through the optical fiber? The noise? A second example: "In contrast, the electron transport through the π -bridge pathway on SDAS relies on

a distinct main-chain modulation mechanism (Fig. 2b)” How can the transport rely on a modulation mechanism? Probably the authors mean that the change of the transport relies on a modulation mechanism.

Response: Thanks to the reviewer for the kind comments. The specific sentence mentioned has now been fully revised for clarity and technical accuracy as follows:

“Building on the insights obtained from the side-chain modulation in SSDA, we next examined a distinct modulation strategy in SDAS, where electron transport necessarily traverses the entire molecular backbone (Fig. 4a).”

“A 635 nm laser was positioned outside the main STM-BJ setup to minimize vibration and electrical noise generated by its cooling system and power-supply fans, and its optical output was delivered to the junction cell through a long optical fiber.”

“In contrast, for SDAS the HOMO is distributed along the main conjugated backbone, and the changes in electron transport occur through a distinct main-chain modulation mechanism operating along the π -bridge (Fig. 2b)”

We have additionally reviewed the manuscript for language issues and grammatical inconsistencies, and the revised version has been polished by a native English speaker to ensure clarity and accuracy.

Comment 11

I did not read the 106 pages long supplementary information in all details. However, I saw also here some major flaws that affect the whole analysis. An example: The calculation of the resistances out of the conductance values on pages S32ff seem to be incorrect. A conductance of $G = 10^{(-3.9)} G_0$ corresponds to about 100 M Ω and not 102 k Ω . All values seem to be totally off. 1 G $_0$ is ~12.9 k Ω , 0.1 G $_0$ (i.e. $\log G/G_0 = -1$) is 129 k Ω . Hence 102 k Ω is in the order of $\log G/G_0 = -0.8$ and not $\log G/G_0 = -3.9$, as is claimed there. The above list of criticisms is by far incomplete. This manuscript has to be rejected. I do not see any chance for a revision to result in an acceptable version, since it contains many unsupported big claims and misconceptions. It is lacking a clear description of the experiment, in particular for the single-anchored and the triple anchored molecules.

Response: Thanks to the reviewer for the kind comments. We fully acknowledge that the resistance conversion in the original Supplementary Information contained a basic numerical mistake. All values should indeed be reported in **M Ω rather than k Ω** , and we have now carefully re-checked and corrected all conductance-to-resistance conversions for the molecular systems.

Importantly, this error affects only the **illustrative simulation section** and does not influence any of the experimental STM-BJ results, conclusions, or interpretations. In response to this and your earlier comments, we have made the following major corrections:

(1) The simulation figure and its associated analysis have been completely removed. Since the simulated equivalent-resistance model was not essential to the core scientific conclusions, removing it eliminates a source of potential confusion.

(2) All relevant text has been revised to prevent any implication that a transistor-like three-terminal operation was achieved. The previous analogy to BJT- or MOS-type behavior has been substantially toned down. We now retain only a brief structural analogy to distinguish the *linear* and *cyclic* configurations qualitatively (**Fig. R12**), without suggesting any functional transistor implementation.

(3) All overstated or potentially misleading descriptions have been deleted. The rewritten text avoids any interpretation relying on the removed simulation.

These revisions are clearly marked in red in the updated manuscript. We sincerely appreciate the reviewer for identifying this error. It allowed us to improve the Supplementary Information and refine the manuscript to maintain technical rigor and clarity.

Reviewer #1

The authors have done a good job revising the manuscript. I am, however, wondering about one thing: The HR-MS (ESI) data look fine for all compounds, while none of the HR-MS (MALDI-TOF) data fit the calculated m/z . What is the reason for this?

Response: We sincerely thank the reviewer for the positive and encouraging comments and for raising this important question. Upon careful comparison of the two high-resolution mass spectrometric techniques, we found a clear and systematic difference in their mass accuracy. As summarized in **Fig. R1**, the average deviation between the experimental and calculated m/z values is 0.9221 for HR-MS (MALDI-TOF), whereas it is only 0.0002 for HR-MS (ESI⁺), indicating that the ESI measurements are significantly more reliable in our case.

Notably, all MALDI-TOF measured m/z values show a consistent negative deviation relative to the calculated values. Further inspection revealed that this systematic underestimation originates from insufficient instrument calibration, likely due to prolonged operation without recalibration. Although the MALDI-TOF spectra in the m/z 300-900 range appear relatively clean and exhibit fewer background or fragment peaks, the absolute mass accuracy is compromised by this calibration drift.

To avoid any potential confusion and to ensure the highest data reliability, **we have therefore decided to remove the HR-MS (MALDI-TOF) data from the Supplementary Information and retain only the HR-MS (ESI⁺) results.**

Fig. R1 | Summary of deviations between experimentally measured m/z values and calculated m/z values obtained from HR-MS (ESI⁺) and HR-MS (MALDI-TOF). Comparison of the deviations between experimentally measured m/z values and calculated m/z values obtained from HR-MS (ESI⁺) and HR-MS (MALDI-TOF) for intermediate 3, DAS, SDA, SDAS, SSDA, and SSDAS. The corresponding average deviations of HR-MS (ESI⁺) and

HR-MS (MALDI-TOF) across all compounds are also shown.

Reviewer #2

The authors have carefully addressed all the review comments and made sufficient revisions to the manuscript. I am pleased to recommend the manuscript for acceptance.

Response: We sincerely thank Reviewer #2 for the positive evaluation and the recommendation for acceptance. The authors greatly appreciate the time and support.

Reviewer #3

The authors have thoroughly revised their manuscript and have removed the most problematic parts of their material (e.g. the simulations which were totally wrong and clarified several unclear aspects. Still, I find the manuscript highly intriguing, misleading and unpublishable. The general impression remains that the work was and is premature and that the whole story is not sufficiently thought through. I am deeply convinced that the scientific work should be done by the authors and not by the reviewers of a manuscript. Throwing tons of data onto the readership and the reviewers (in the very lengthy rebuttal letter and revised supplementary information) and letting them fish out the important parts is simply impolite and does not convince me at all. Rather, it indicates that the authors do not really know which parts are really important for conveying their message. Below I will give a selection of major concerns that have not been settled.

Response: We thank Reviewer #3 for the time and effort devoted to evaluating the revised manuscript. We respectfully acknowledge the reviewer's opinion regarding the scope and maturity of the work. While we believe that the revised manuscript now presents a coherent and well-supported scientific message, we appreciate the reviewer's candid feedback and understand that they remain unconvinced.

Comment 1

About original comment 2

The author's reply fixes the strictly wrong statements in the original manuscript but does not address the main problem: How can molecules be useful that require steady irradiation?

Response: We thank the reviewer for raising this important question. Photochromic molecules are commonly classified into P-type and T-type, as well as positive and negative photochromes. (*Chem. Soc. Rev.* **2025**, *54*, 7347-7376.; *Molecular Photoswitches: Chemistry, Properties, and*

Applications, Ed. Z. L. Pianowski, Wiley-VCH, 2023.)

The requirement for steady irradiation is inherently linked to the operating mechanism of T-type photochromes and reflects their fundamental photochemical behavior. T-type photoswitches (*e.g.* azobenzene, spiropyran, donor-acceptor Stenhouse adducts) exhibit light-induced isomerization to one direction, where the reversed isomerization occurs either through light-activation with longer wavelength or through thermal-relaxation. **First**, steady irradiation enables continuous and reversible access to a well-defined photostationary state, which is compatible with optoelectronic operating conditions involving continuous illumination and supports real-time signal processing, as exemplified by reported photo-synaptic systems (*Commun. Mater.* **2025**, *6*, 11) and photo-controlled logic devices (*Adv. Sci.* **2023**, *10*, 2207443). **Second**, the automatic thermal recovery of T-type systems makes them particularly suitable for transient or adaptive functions without the need for a secondary stimulus, as demonstrated in adaptive photoresponsive systems (*Sci. Adv.* **2024**, *10*, eads2217).

It should be noted, however, that these the above mentioned three studies mainly address the collective behavior of T-type photochromes at the ensemble level, while the present work is devoted to revealing the underlying molecular-level chemical mechanisms of T-type switching, thereby providing fundamental insight beyond ensemble-averaged effects. In this study, continuous irradiation is used to enrich the *cyclic* isomer of DASAs, which helps the determination of molecular conductance.

Comment 2

About original comment 3

In their response the authors agree that they explore not more than two conductance pathways. Still, in their revised material they insist in claiming multiple pathways and multiple anchoring. The latter could be fixed by writing “multiple anchoring possibilities”, what would be not new though.

Response: We thank the reviewer for this careful and constructive suggestion.

(1) Revision of “multiple” terminology

In the revised manuscript, we have clarified and consistently refined our terminology to accurately reflect the scope of the experimental evidence. Specifically, “**multiple conductive pathways**” has been revised to “**two conductive pathways**,” and “**multiple anchoring**” has been replaced by “**two/three anchoring configurations**.”

To ensure consistency throughout the manuscript, the title has been revised from:

“Photogated Multiple Conductive Pathways of Donor-Acceptor Stenhouse Adducts in Single-Molecule Junctions”

to:

“Photogated Two Conductive Pathways of Donor-Acceptor Stenhouse Adducts in Single-Molecule Junctions.”

These revisions have been implemented consistently across the main text and Supplementary Information.

(2) Clarification of the novelty of the present work

We agree with the reviewer that the concept of multiple anchoring itself is not new and has been explored in previous studies, for example in multi-anchored perovskite systems (*Nat. Commun.* **2019**, *10*, 5458). and in mechanically tuned single-molecule junctions (*Angew. Chem. Int. Ed.* **2023**, *62*, e202302693.). The novelty of the present work does not arise from the existence of multiple anchoring configurations.

Instead, the key advance of this study is the demonstration that **two distinct conductive pathways can be reversibly controlled by light at the single-molecule level**. This photogated control of conductance pathways, rather than static multiple anchoring, constitutes the central contribution of the work.

Comment 3

About original comment 4

The answer is not to the point. Fig. 1 a shows three electrodes to the molecule. This suggests that the same molecule would be contacted by three electrodes simultaneously This is simply wrong. The figure is more than misleading. It is pretending another experiment than what is done. Furthermore, throughout the whole manuscript it is not clearly stated what they are doing: I understood the experiment as follows: They have a three-legged molecule and believe that it is contacted to the bottom electrode by one of these legs and with the tip (that may have several protrusions) two of these legs. Is this a correct interpretation? The comparison with the piezoelectric drive in Fig. 1b is additionally misleading, but probably not wrong. The major problem of the misleading figure cannot be healed by rewriting the figure caption. Additional questions. What is the meaning of the blueish or transparent bubbles?

Response: We thank the reviewer for this detailed and technically important comment, which prompted us to further clarify several aspects of **Fig. 1a, b** and the associated description.

(1) Clarification of the electrode-molecule contact geometry.

We agree that the original schematic in Fig. 1a could be misinterpreted as implying simultaneous contact of a single molecule with three electrodes. In the revised figure (**Fig. R2**), the schematic has been redesigned to explicitly reflect a *sequential and dynamic contact process* within a conventional two-electrode STM-BJ configuration.

(2) Clarification of the electrical circuit representation.

In the original figure, the circuit layout could be misread as a parallel connection. This has now been corrected by redrawing the circuitry as *two independent series circuits*, corresponding to distinct measurement configurations (**Fig. R2**).

(3) Revision of Fig. 1b

In the revised **Fig. 1b**, all electrode representations have been removed. The schematic now focuses exclusively on the molecular structure and the two distinct conductive pathways within the molecule.

(4) Clarification of the solvent environment (“bubble”).

We acknowledge that the “bubble” in the original schematic was insufficiently defined. In the revised figure, this element is explicitly defined as a *solvent droplet* with a finite refractive index (**Fig. R2**).

(5) Origin of the blue coloration.

The blue coloration arises from the intrinsic absorption of all five DASAs studied in this work when dissolved in TCB. To clarify this point, **Fig. R2** shows photographs of the corresponding solutions for *linear* SSDA and *cyclic* SSDA.

Fig. R2 | Summary of revisions to Fig. 1a, b.

Comment 4

Furthermore, the figure caption is contradicting what is shown and what the authors replied in an earlier comment: If all measurements were done in the dark, what's about 635 nm irradiation?

Response: Thanks to the reviewer for the kind comments. Specifically, all measurements corresponding to the *linear state* were performed **under dark conditions**, whereas all measurements corresponding to the *cyclic state* were performed **under 635 nm laser irradiation**.

We have double-checked the entire manuscript and the Supplementary Information to ensure consistency. In the Supplementary Information, the original sentence,

"All measurements were carried out in the dark at 298 K."

has been revised to:

"All measurements were carried out at 298 K, with conductance measurements performed either in the dark or under 635 nm laser irradiation, as specified in the main text."

Comment 5

About original comment 5

Also in their revised material the authors claim that the HOMO would be delocalized over the whole length (see e.g. response to reviewer 2, comment 10). This is wrong (or at least not supported by evidence). There is no doubt that the HOMO might be delocalized in some molecules (the long list of examples provided by the authors). But this is not to the point. For their argument it would be important to show that in this particular case studied here the HOMO is indeed delocalized over the whole length)

Response: Thanks to the reviewer for the kind comments. To provide a more accurate description of the HOMO distribution, all statements referring to HOMO delocalization over the “**whole molecular length**” have been revised. The HOMO is now consistently described as being primarily distributed over the **triene π -bridge**, as supported by the calculated HOMO isosurfaces shown in **Fig. R3**.

In the main text, the original description:

“In its linear state, the HOMO spans from donor to acceptor across the triene π -bridge, indicating a well-coupled transport path. Following the light-triggered linear-to-cyclic isomerization, the HOMO becomes unevenly localized on the donor side, with the orbital abruptly truncated at the newly formed cyclopentenone.”

Has been revised to:

“In its linear state, the HOMO shows significant contribution over the triene π -bridge. Upon light-induced cyclization, the disruption of conjugation by the cyclopentenone unit leads to a more localized HOMO distribution on the donor side.”

The original description:

“In contrast, photoisomerization directly reshapes the π -bridge of SDAS. For the linear isomer, the HOMO is delocalized across the donor and the extended π -system, supporting a continuous channel.”

Has been revised to:

“In contrast, photoisomerization directly reshapes the π -bridge of SDAS. In the linear isomer, the HOMO shows significant contribution over the extended π -system.”

Fig. R3 | Calculated HOMO distributions of *linear* and *cyclic* SDAS.

Comment 6

About original manuscript 6

The answer does not reply to the question., Probably my original comment was not clear. I meant: “For the single-anchored molecules: How do the authors know where the top tip contacts the molecule?”

Response: Thanks to the reviewer for the kind comments. In principle, Au electrodes can interact with several atoms or functional groups in organic molecules, including S, C, N, and O. Among these, Au-S interactions are known to exhibit strong binding affinity, making thiomethyl groups the dominant anchoring motif under STM-BJ conditions (*Nat. Rev. Mater.* **2016**, *1*, 16002; *Nat. Commun.* **2025**, *16*, 7692; *J. Am. Chem. Soc.* **2012**, *134*, 19425-19431; *J. Am. Chem. Soc.* **2017**, *139*, 14845-14848).

For single-anchored molecules, in principle, a molecular junction could be formed between one thiomethyl anchor and a secondary, weaker interaction involving C, N, or O.

In the present work, as stated in our response to Comment 6 in the previous revision, we could have completely removed all data related to the SDA and DAS molecules. However, the **sole reason** for including these two molecules is to exclude and verify that the conductance features observed for the main-chain and side-chain designs are reliable and do not originate from other possible anchoring scenarios that could lead to misinterpretation. Specifically,

“the single-anchored molecules DAS and SDA were specifically designed and measured as control experiments to determine whether unintended anchoring could contribute to the conductance features observed in SSDA and SDAS.

Under identical STM-BJ conditions, including three bias voltages (100, 300 and 500 mV) and both irradiation states, neither DAS nor SDA exhibited conductance features similar to those of SSDA, SDAS and SSDAS (Supplementary Figs. 24-30). These results demonstrate that non-thiomethyl anchoring does not generate the conductance signatures associated with donor or π -bridge pathways. A detailed discussion of unconventional anchoring is provided in Section 6.2 of the SI.”

Comment 7

About original comment 7

“About “V shape” dependence of the conductance on the bias voltage: There are only three data points: 100 mV, 300 mV, 500 mV. It is exaggerated to talk about V shape. It seems to be non-monotonous, also the change is smaller than the width of the distributions. Is this really significant? The same question for the minor change of the Flicker noise exponent from 1.5 to 1.6. I doubt that when repeating the experiments on another day, with another tip one would be able to reproduce the peak positions of the conductance histograms or the noise characteristics with the claimed precision.

Response: Thanks to the reviewer for the kind comments.

(1) Bias-dependent conductance. We appreciate the reviewer’s concern. Most single-molecule conductance studies report results measured at only one bias voltage, as summarized in the review on photoswitchable junctions (Chem. Commun. 2023, 59, 12685). Apart from a few specialized investigations focusing on bias-dependent effects (J. Am. Chem. Soc. 2023, 145, 21679-21686), systematic multi-bias analyses remain uncommon. Considering this, we have removed the term “V shape” and now describe the bias dependence simply as an objective experimental observation. For clarity and conciseness, the main text now focuses on the data collected at 300 mV. “

New comment: Other authors do not claim a V dependence of the conductance at all. Hence, these author do not need to provide evidence. Still, I appreciate the authors removed this overinterpretation of the data.

(2) Reproducibility. To ensure that the observed conductance features are reliable, each molecule was tested under five independent and orthogonal conditions, including three bias voltages under irradiation and in the dark as well as three independent datasets at 300 mV (in addition to the initial 300 mV measurement, we performed two additional light-off/light-on measurement cycles, resulting in three complete datasets at 300 mV). The resulting conductance histograms reproduce highly consistent conductance features across all conditions. (Fig. R8-R10 and Supplementary Figs. 32-45). The orthogonal repetition of the measurements

substantially reinforces the reproducibility of the observed features.

New comment: This answer is not to the point. I did not ask about the reproducibility of the conductance measurements. (I raised this aspect of reproducibility only for the Flicker noise measurements My question about the statistical significance remains unanswered. If the distributions are wider than the separation of their main values, some analysis of the statistical significance has to be provided.

Response: Thanks to the reviewer for the kind comments.

(1) Reproducibility of flicker noise measurements.

To directly address this point, we performed two additional independent noise measurements under identical conditions, resulting in **three independent flicker noise datasets** for both *linear* and *cyclic* SSDA. The noise power scaling exponent (G) was extracted in each case using the instrument-provided analysis software based on the fitted density contours (Fig. R4). For *linear* SSDA, the extracted exponents are 1.51, 1.47, and 1.49 (average = 1.49), while for *cyclic* SSDA they are 1.61, 1.64, and 1.58 (average = 1.61). These values are reported to demonstrate measurement reproducibility. The corresponding **averages** and **standard deviations** are summarized in **Fig. R4**, demonstrating good reproducibility of the noise characteristics across independent measurements.

(2) Statistical significance and interpretation.

As reflected in the revised manuscript, the previous statement suggesting that “*cyclic SSDA is slightly more inclined toward through-space transport than linear SSDA*” has been replaced with a non-overinterpreting description, stating that “*the exponents in the range of 1.5-1.6 observed experimentally reflect a combined contribution of the two mechanisms.*” In the newest revised version, we further emphasized at the end of the sentence that **these values do not exhibit significant differences between *linear* and *cyclic* SSDA.** The revised sentence:

“Therefore, the intermediate exponents (1.5-1.6) observed experimentally reflect a combined contribution of the two mechanisms, which do not exhibit significant differences between linear and cyclic SSDA.”

Fig. R4 | Flicker noise power spectral density (PSD) analysis of SSDA in the *linear* and *cyclic* states. Two-dimensional histograms of noise power versus conductance for *linear* SSDA (top row) and *cyclic* SSDA (bottom row), obtained from **three independent parallel measurements** (First, Second and Third). The solid contours indicate the density distribution, from which the noise power scaling exponent (G) is extracted for each dataset. The right panel summarizes the **average noise power scaling exponent** for the *linear* and *cyclic* states, with **error bars representing the standard deviation** across the three measurements.

Comment 8

About original comment 8

I appreciate the revision. Still, the last sentence is overselling. The term “multiple” suggests “two or more” and is therefore overselling. The authors study the possibility of exactly two pathways and not more.

Response: Thanks to the reviewer for the kind comments. As in Comment 2, all references to “multiple” pathways or anchoring have been revised to explicitly refer to **two conductive pathways** and **two/three anchoring configurations** throughout the manuscript and the Supplementary Information.

Comment 9

About original comment 9

I appreciate the revision which reduces the amount of misleading and overselling statements considerably, Still, there is simply NO analogy to any transistor, as the authors agree on in their response. Instead, in the revised abstract they suggest this analogy again. So, this issue is not solved.

Response: Thanks to the reviewer for the kind comments. As suggested, the transistor-related analogy have been removed from the abstract. The revised abstract now focuses exclusively on the experimentally observed photoinduced transition between a mixed (through-bond/through-space) transport mechanism and a through-space-dominated transport mechanism.

The original description:

*“This photoinduced shift from π -bridge hybrid (through-bond/through-space mixed) transport in the linear state to cyclopentenone-driven through-space transport in the cyclic state renders the two isomers **structurally analogous** to bipolar junction transistor (BJT)-like and metal-oxide-semiconductor (MOS)-like configurations.”*

Has been revised to:

“The π -bridge hybrid (through-bond/through-space mixed) transport in the linear state shifts to cyclopentenone-driven through-space transport in the cyclic state under light irradiation.”

Reviewer #1

The authors have removed the MALDI MS results that seemed to be wrongly calibrated. I therefore recommend acceptance of the manuscript.

Response: We sincerely thank the reviewer for the positive and encouraging comments and for raising this important question.

Reviewer #3

Summarizing, the authors admitted that some of their original claims were wrong or exaggerated and turned them down. Still, after two rounds of revision major flaws remain (transistor comparison, claim of switching of transport character). I repeat and extend one of my remarks of my previous (2nd) report: “The general impression remains that the work was and is premature and that the whole story is not sufficiently thought through. I am deeply convinced that the scientific work should be done by the authors PRIOR TO SUBMISSION and not by the reviewers of a manuscript.”

If the transistor part is removed from the manuscript and the other incorrect statements are corrected, the manuscript, I do not object against publication. If the remaining parts are substantial enough to warrant publication in Nature Communications is an editorial decision.

Response: We thank the reviewer for the careful evaluation. In response to the remaining concerns, we have (1) completely removed all **transistor-related** comparisons, figures, and discussions from the manuscript and Supplementary Information; (2) revised descriptions of **transport mechanisms** to avoid claims of complete switching; and (3) clarified the **junction configuration** and **anchoring description**, explicitly stating the two-terminal nature of the junctions and the uncertainty in atomic-scale contact geometry.

Comment 1

For instance, Fig. 1a now shows that they do study single-molecule junctions in two-contact geometry (instead of three as in the original Fig. 1a). Interestingly, by revising Fig. 1a the authors have also changed the initially assumed contact geometry by inverting the orientation of the molecule in the junction. I agree that the current orientation fits better to the interpretation of the stretching curves (short junctions corresponding to donor pathway, long junctions to the pi bridge configuration).

Response: We thank the reviewer for the positive assessment.

Comment 2

Unfortunately, in some figure (right-most illustration of Fig. 1d) they still show three electrodes contacting the molecule. According to their response to my previous comments, they agree that the molecule is only coupled to two electrodes simultaneously. Still, this is not explicitly stated in the manuscript. Furthermore, they authors still use the term “multi-anchored” in Fig. 1d and “triple-anchored” at several places in the text, although there is no triple-anchored junction. The molecule has three anchoring sites, yes, but only two of them are used simultaneously. The

switching between the donor and the pi-bridge configuration is done by stretching, i.e., one contact to one anchoring site breaks and a contact to another one is formed. This is evidenced by the two-plateau shape of the stretching curves and the data is consistent with this interpretation. The two conduction pathways share one leg that remains stable and one that is swapped. Also this simple fact should be written down explicitly.

Response: We thank the reviewer for this careful and constructive suggestion.

1. All electrode schematics in **Fig. 1d** that could imply simultaneous three-electrode contacts have been removed (see revised **Fig. R1d**).
2. The terms “Single-anchored” and “Multi-anchored” in the original figure have been replaced with the more explicit descriptions “Single anchoring site”, “Two anchoring sites”, and “Three anchoring sites”.

*“d Chemical structures of DASAs with a **single anchoring site** (DAS and SDA), **two anchoring sites** (SDAS and SSDA), and **three anchoring sites** (SSDAS)”*

3. The caption of **Fig. 1d** has been revised accordingly by using “anchoring site” terminology throughout, and the following clarification has been added at the end of the caption: *“Molecules with **three anchoring sites** enabling alternative **two-terminal junction configurations**.”*

4. The anchoring-leg exchange mechanism has been explicitly described in the main text: the two conductive pathways share one common anchoring leg that remains stable, while the other anchoring leg is exchanged during mechanical stretching.

*“Although the molecule contains **three anchoring sites**, at any given time only two are simultaneously coupled to the electrodes to form a two-terminal junction. **The transition between the donor and the π -bridge pathways occurs during mechanical stretching, where one anchoring contact remains stable while the other anchoring contact is exchanged.**”*

5. Throughout the manuscript, the Supplementary Information, and all figures and figure captions, the terms “single-anchored junction”, “double-anchored junction”, “triple-anchored junction”, and “multi-anchored junction” were consistently replaced with structure-based descriptions, specifically “molecules with multiple anchoring sites” and “alternative two-terminal configurations”, as appropriate.

Fig. R1 | The revised Fig. 1

Comment 3

The photogating mechanism is a change of the molecular conformation that goes along with a change of the electronic structure. Thereby the conductance of the long pi bridge pathway is modified by more than a decade and the one of the short donor is only slightly modified (by around 0.2 decades and hence less than the width of the corresponding histogram peak). This is all nicely evidenced by the photochemical and transport investigations. The fact that both pathways are affected is not surprising at all, but still a nice functionality and has to the best of my knowledge not been reported before. Still, selling it as “simultaneous photogating of two pathways” is overselling: One pathway is affected, the other one is mainly unaffected.

Response: We thank the reviewer for the positive assessment and in particular for the recognition that this represents “a nice functionality and, to the best of the reviewer’s knowledge, has not been reported before.”

We agree that the magnitude of the photogating effect is markedly different for the two conductive pathways. Upon photoisomerization, the donor pathway exhibits only a modest conductance increase of approximately 0.2 orders of magnitude for SSDA and 0.26 orders of magnitude for SSDAS. Although this change is relatively small compared with that of the π -bridge pathway, it is reproducible across multiple independent measurements (**Fig. 3**, **Supplementary Fig. 35**, **Fig. 5**, and **Supplementary Fig. 45**) and is consistently supported by the calculated transmission spectra (**Fig. 3f** and **Supplementary Fig. 49**).

Comment 4

Wordings like "... their modular structure that allows for triple-anchor configuration" and the comparison to transistors suggests that there would be three contacts simultaneously to one molecule. This is not the case and gives a completely wrong interpretation of the functionality. The worst is Fig. 6c, which draws again three terminals to the molecule. The described experimental evidence has nothing to do with transistor action at all. A transistor is a three-terminal device. The molecule studied here (the "triple anchored" one) is a device that has two different possibilities for a two-terminal configuration (the donor pathway and the pi bridge pathway) by keeping one anchoring group the same and changing the other one. The pi bridge pathway in both conformations (linear and cyclic) is very resistive, the donor pathway has a lower resistance in both conformations (also not surprising since the pi bridge is much longer). The resistance change to the pi bridge pathway upon photoisomerization is more pronounced than the one to the donor pathway (also not surprising since the conformational change takes place in the pi bridge part). The general situation remains the same: Highly resistive pi bridge pathway and low-resistive donor pathway. The similarity of both situations is also supported by the noise measurements: The difference between the linear and the cyclic case are not significant. There is absolutely no evidence to claim a transistor action at all and even less to claim a switching to another type of transistor. The comparison with transistors must be removed from the article, since it is scientific nonsense. Btw, assigning specific resistances to the individual arms is also highly suggestive and indicating the resistances with four digits is simply not justified by the data. And another remark: I am not sure if the figure caption of Fig. 6 a and b really fit to what is shown. I do not see red and blue arrows and I also do not know what a negative and a positive transport channel would be.

Response: Thanks to the reviewer for the kind comments. We thank the reviewer for raising this important point. We recognize that the previous use of **potential**, **-like**, and **structural analogy** language, together with the presentation of Fig. 6, could nonetheless give rise to an unintended interpretation of three-terminal or transistor-like functionality.

To avoid any possible ambiguity, **Fig. 6 has been removed in its entirety, and all transistor-**

related descriptions and comparisons have been eliminated from both the revised manuscript and the Supplementary Information.

Comment 5

Furthermore, claiming “The π -bridge hybrid (through-bond/through-space mixed) transport in the linear state shifts to cyclopentenone-driven through-space transport in the cyclic state under light irradiation.”) is also wrong. Both have mixed character as shown by the noise data.

Response: We thank the reviewer for this comment. For the pi-bridge pathway, the flicker noise exponents increase from **1.54** to **1.94** for **SDAS** and from **1.61** to **1.92** for **SSDAS** upon photoisomerization, indicating that both the *linear* and *cyclic* states retain mixed through-bond and through-space transport character, rather than **purely through-space transport (ideal value of 2.00)**.

To reflect this more accurately, we have revised the manuscript to replace statements implying a complete transition with more precise descriptions, such as **“exhibits an increased through-space contribution”** and **“shows a pronounced shift toward through-space-assisted transport.”**

The original statement:

*“The π -bridge hybrid (through-bond/through-space mixed) transport in the linear state **shifts to** cyclopentenone-driven through-space transport in the cyclic state under light irradiation.”*

Has been revised to:

*“The π -bridge transport in the linear state exhibits mixed through-bond and through-space character, while photoisomerization leads to an **increased through-space contribution** in the cyclic state driven by cyclopentenone formation.”*

The original statement:

*“Specifically, the linear SDAS exhibits noise characteristics of a mixed-mode serial junction consistent with a ‘resistive’ through-bond path supplemented by a ‘capacitive’ **through-space coupling channel**. In contrast, the cyclic SDAS behaves as a **through-space dominated junction**, wherein the conductance response resembles that of a **pure capacitor**, with suppressed direct tunneling and enhanced electrostatic coupling.”*

Has been revised to:

“Specifically, the linear SDAS exhibits noise characteristics consistent with mixed through-bond and through-space transport, where a dominant through-bond pathway is supplemented by through-space coupling. In contrast, photoisomerization to the cyclic form leads to a

*substantial suppression of conductance, accompanied by an **enhanced contribution from through-space interactions and weakened through-bond transport.***”

The original statement:

*“(2) the π -bridge pathway is achieved on SDAS, where the electrons traverse the entire molecule (i.e. donor, π -bridge and acceptor), the formation/deformation of the π -bridge on the main chain switches the conjugated structure between **hybrid** (through-bond/through-space mixed) transportation in linear and **through-space transportation in cyclic**, which generates a conductance variation within 0.62 orders of magnitude.”*

Has been revised to:

*“(2) The π -bridge pathway is achieved in SDAS, where electrons traverse the entire molecule (i.e., donor, π -bridge, and acceptor). Photoisomerization induces formation and deformation of the π -bridge along the main chain, leading to a significant redistribution between through-bond and through-space contributions: the linear state exhibits mixed through-bond/through-space transport, while the cyclic state shows an **enhanced contribution from through-space transport**, resulting in a conductance variation of 0.62 orders of magnitude.”*

Comment 6

About original comment 6: Within the lengthy response, the only sentence which is to the point is: ” For single-anchored molecules, in principle, a molecular junction could be formed between one thiomethyl anchor and a secondary, weaker interaction involving C, N, or O. ”. Or in other words: The contact geometry is not really known. Although this is a minor point, it should be stated somewhere.

Response: We thank the reviewer for this clarification. As suggested, we have now explicitly stated in the main text, at the point where the single-anchored DAS and SDA molecules are introduced, that the atomic-scale contact geometry in STM-BJ measurements cannot be uniquely determined. We further clarified that, while weaker secondary interactions involving C, N, or O atoms may in principle contribute to junction formation, the control experiments indicate that such non-thiomethyl anchoring does not play a dominant role in the conductance features discussed.

The original statement:

*“Moreover, to **rule out** the possibility of anchoring through atoms other than the thiomethyl group, two molecules with a single thiomethyl anchoring site (DAS and SDA) were designed (Fig. 1d). Under the same conditions with multiple biases, neither of these two molecules exhibit conductance features similar to those of SSDA and SDAS (the issue of unconventional*

anchoring is discussed in detail in section 6.2 of the SI, Supplementary Figs. 24-30). This implies that in the two sets of tests mentioned above, the possibility of non-thiomethyl anchoring could be ruled out.”

Has been revised to:

*“Moreover, to assess the possibility of anchoring through atoms other than the thiomethyl group, two molecules with a single thiomethyl anchoring site (DAS and SDA) were designed (Fig. 1d). Under the same conditions with multiple biases, neither of these two molecules exhibit conductance features similar to those of SSDA and SDAS (the issue of unconventional anchoring is discussed in detail in section 6.2 of the SI, Supplementary Figs. 24-30). **While the exact atomic-scale contact geometry in STM-BJ measurements cannot be uniquely determined, these control experiments indicate that contributions from non-thiomethyl anchoring, including possible weaker interactions involving C, N, or O atoms, if present, do not play a dominant role in the conductance features discussed above.**”*